# Dynamics of ribosomes and release factors during translation termination in *E. coli*

**Sarah Adio\*, Heena Sharma, Tamara Senyushkina, Prajwal Karki, Cristina Maracci, Ingo Wohlgemuth, Wolf Holtkamp, Frank Peske, Marina V Rodnina\***

Department of Physical Biochemistry, Max Planck Institute for Biophysical Chemistry, Göttingen, Germany

**Abstract** Release factors RF1 and RF2 promote hydrolysis of peptidyl-tRNA during translation termination. The GTPase RF3 promotes recycling of RF1 and RF2. Using single molecule FRET and biochemical assays, we show that ribosome termination complexes that carry two factors, RF1–RF3 or RF2–RF3, are dynamic and fluctuate between non-rotated and rotated states, whereas each factor alone has its distinct signature on ribosome dynamics and conformation. Dissociation of RF1 depends on peptide release and the presence of RF3, whereas RF2 can dissociate spontaneously. RF3 binds in the GTP-bound state and can rapidly dissociate without GTP hydrolysis from termination complex carrying RF1. In the absence of RF1, RF3 is stalled on ribosomes if GTP hydrolysis is blocked. Our data suggest how the assembly of the ribosome–RF1–RF3–GTP complex, peptide release, and ribosome fluctuations promote termination of protein synthesis and recycling of the release factors.
DOI: https://doi.org/10.7554/eLife.34252.001

**\*For correspondence:**
Sarah.Adio@mpibpc.mpg.de (SA);
rodnina@mpibpc.mpg.de (MVR)

**Competing interests:** The authors declare that no competing interests exist.

## Introduction

Termination of protein synthesis occurs when a translating ribosome encounters one of the three universally conserved stop codons UAA, UAG or UGA. In bacteria, the release of the nascent peptide is promoted by release factors RF1 and RF2 which recognize the stop codons in the A site and hydrolyze the ester bond in the peptidyl-tRNA bound to the P site, allowing the nascent peptide to leave the ribosome through the polypeptide exit tunnel (*Dunkle and Cate, 2010*; *Nakamura et al., 1996*). RF1 and RF2 bind to the ribosome in the space between the small and large ribosomal subunits. RF1 and RF2 differ in their stop codon specificity: RF1 utilizes a conserved PET motif to recognize UAG and UAA codons, whereas RF2 uses an SPF motif to recognize UGA and UAA codons. Both RF1 and RF2 have a universally conserved GGQ motif which promotes the catalysis of peptidyl-tRNA hydrolysis (*Seit-Nebi et al., 2001*); mutations of the GGQ motif to GAQ or GGA inhibit peptide release (*Frolova et al., 1999*; *Mora et al., 2003*; *Shaw and Green, 2007*; *Zavialov et al., 2002*). After peptide release, RF1 and RF2 dissociate from the post-termination complex to allow for the next steps of translation. The dissociation is accelerated by RF3, a translational GTPase that binds and hydrolyses GTP in the course of termination (*Freistroffer et al., 1997*; *Koutmou et al., 2014*; *Zavialov et al., 2002*). In addition to canonical termination, RF2 takes part in non-canonical termination events such as post-peptidyl transfer quality control (*Zaher and Green, 2009*) and ribosome rescue on truncated mRNAs (*Kurita et al., 2014*).

There are two different models concerning the sequence of events during termination, including the timing of peptide release, the order of RF1, RF2 and RF3 binding and dissociation, and the role of nucleotide exchange in RF3 and GTP hydrolysis. The first model of translation termination was proposed by Ehrenberg and colleagues (*Zavialov et al., 2001*; *Zavialov et al., 2002*). Based on

**eLife digest** Inside cells, molecular machines called ribosomes make proteins using messenger RNA as a template. However, the template contains more than just the information needed to create the protein. A 'stop codon' in the mRNA marks where the ribosome should stop. When this is reached a group of proteins called release factors removes the newly made protein from the ribosome.

Bacteria typically have three types of release factors. RF1 and RF2 recognize the stop codon, and RF3 helps to release RF1 or RF2 from the ribosome so that it can be recycled to produce another protein. It was not fully understood how the release factors interact with the ribosome and how this terminates protein synthesis.

Adio et al. used TIRF microscopy to study individual ribosomes from the commonly studied bacteria species *Escherichia coli*. This technique allows researchers to monitor movements of the ribosome and record how release factors bind to it. The results of the experiments performed by Adio et al. show that although RF1 and RF2 are very similar to each other, they interact with the ribosome in different ways. In addition, only RF1 relies upon RF3 to release it from the ribosome; RF2 can release itself. RF3 releases RF1 by forcing the ribosome to change shape. RF3 then uses energy produced by the breakdown of a molecule called GTP to help release itself from the ribosome.

Most importantly, the findings presented by Adio et al. highlight that the movements of ribosomes and release factors during termination are only loosely coupled rather than occur in a set order. Other molecular machines are likely to work in a similar way. The results could also help us to understand the molecular basis of several human diseases, such as Duchenne muscular dystrophy and cystic fibrosis, that result from ribosomes not recognizing stop codons in the mRNA.

DOI: https://doi.org/10.7554/eLife.34252.002

nitrocellulose filtration experiments, the authors reported that free RF3 has a much higher affinity for GDP ($K_d$ = 5.5 nM) than for GTP ($K_d$ = 2.5 µM) or GDPNP ($K_d$ = 8.5 µM) (*Zavialov et al., 2001*), which would imply that at cellular GTP/GDP concentrations RF3 is expected to be predominantly in the GDP form. The exchange of GDP for GTP occurs only when RF3–GDP binds to the ribosome in complex with RF1 or RF2 (*Zavialov et al., 2001*). In the absence of the nucleotide, RF3-dependent RF1/2 recycling is slow, which has been interpreted as an indication for a high-affinity complex of apo-RF3 to the ribosome–RF1/2 complex (*Zavialov et al., 2001*). Furthermore, because RF3-dependent turnover GTPase activity was stimulated by peptidyl-tRNA hydrolysis, the authors suggested that RF3 binds to the ribosome–RF1 complex only after the peptide is released. Based on these results, Ehrenberg et al. suggested the following sequential mechanism of termination: RF1/RF2 bind to the ribosome and hydrolyze peptidyl-tRNA, allowing RF3–GDP to enter the ribosome occupied by RF1 or RF2 to form an unstable encounter complex. Dissociation of GDP leads to a stable high-affinity complex with RF3 in the nucleotide-free state. The subsequent binding of GTP by RF3 promotes RF1/RF2 dissociation. In the final step, RF3 hydrolyses GTP and as a result dissociates from the ribosome (*Zavialov et al., 2001*; *Zavialov et al., 2002*).

An alternative model was proposed based on the kinetic and thermodynamic analysis of GTP/GDP binding to RF3 by ensemble kinetics and equilibrium methods. The results of those experiments indicated that the affinity of RF3 to GDP and GTP is on the same order of magnitude (5 nM and 20 nM, respectively [*Koutmou et al., 2014*; *Peske et al., 2014*]). As the cellular GTP concentration is at least 10 times higher than the GDP concentration (*Bennett et al., 2009*), these affinities imply that nucleotide exchange in RF3 can occur spontaneously, off the ribosome, and thus RF3 could enter the ribosome in either the GTP- or GDP-bound form. Consistent with previous findings (*Zavialov et al., 2001*; *Zavialov et al., 2002*), ribosome–RF1 or ribosome–RF2 complexes accelerate nucleotide exchange in RF3 (*Koutmou et al., 2014*; *Peske et al., 2014*); however, this effect is independent of peptide release, because also a catalytically inactive RF2 mutant activates nucleotide exchange in RF3 (*Peske et al., 2014*; *Zavialov et al., 2002*). Binding of GTP to RF3 in complex with the ribosome and RF2 is rapid (130 s$^{-1}$) (*Peske et al., 2014*), and thus the lifetime of the apo-RF3 state would be too short to assume a tentative physiological role. Peptide release results in the

stabilization of the RF3–GTP–ribosome complex, thereby promoting the dissociation of RF1/2, followed by GTP hydrolysis and dissociation of RF3–GDP from the ribosome (*Peske et al., 2014*).

Efficient translation termination not only requires the coordinated action of the release factors, but also entails conformational dynamics of the factors and the ribosome. The key conformational motions of the ribosome during termination and in general in all phases of translation include the rotation of ribosomal subunits relative to each other, the swiveling motion of the body and head domains of the small ribosomal subunit, the movement of the ribosomal protein L1 toward or away from the E-site tRNA, and the movement of tRNAs between classic and hybrid conformation. These motions are loosely coupled and gated by ligands of the ribosome such as translation factors and tRNAs (*Adio et al., 2015*; *Chen et al., 2011*; *Cornish et al., 2008*; *Horan and Noller, 2007*; *Sharma et al., 2016*; *Shi and Joseph, 2016*; *Valle et al., 2003*; *Wasserman et al., 2016*). Crystal structures show that termination complexes with RF1 or RF2 are predominantly in the non-rotated (N) state. The P-site tRNA in the complexes is in the classical state and the L1 stalk in an open conformation (*Jin et al., 2010*; *Korostelev et al., 2008*; *Laurberg et al., 2008*; *Weixlbaumer et al., 2008*). A single molecule fluorescence resonance energy transfer (smFRET) study showed that binding of RF1 to termination complexes stabilizes the open conformation of the L1 stalk, whereas in the absence of RF1 termination complexes make reversible transitions between the open and closed state (*Sternberg et al., 2009*); the rotation of the ribosomal subunits was not investigated directly in that study. The high sequence similarity between RF1 and RF2 suggests that the two factors interact with the ribosome in the same manner and promote peptide release by a similar mechanism (*Freistroffer et al., 1997*; *Zavialov et al., 2001*). However, structures of RF1 or RF2 bound to termination complexes reveal differences regarding the interaction with the L11 region of the 50S subunit (*Korostelev et al., 2008*; *Laurberg et al., 2008*; *Petry et al., 2005*; *Pierson et al., 2016*; *Rawat et al., 2006*; *Rawat et al., 2003*; *Weixlbaumer et al., 2008*). Thus, it is not clear whether RF1 and RF2 follow the same mechanism and whether they respond in the same way to the recruitment of RF3 to termination complexes.

In the absence of RF1/RF2, binding of RF3 with a non-hydrolyzable GTP analog to the ribosomes where the nascent peptide has been released induces formation of the rotated (R) state of the ribosome, with the tRNA in the P/E hybrid state and the closed conformation of the L1 stalk (*Gao et al., 2007*; *Jin et al., 2011*; *Zhou et al., 2012*). A very similar effect of RF3–GDPNP was found by smFRET (*Sternberg et al., 2009*). However, it is much less clear what happens when RF1/RF2 and RF3 bind to the ribosome together. Modeling of the atomic structures of RF1 and RF2 into the cryo-EM structure of RF3-bound post-release complex suggests that the RF3-induced ribosome rearrangements break the interactions of RF1/RF2 with both the decoding center and the L11 region of the ribosome, leading to the release of RF1/RF2 (*Gao et al., 2007*). In this model, stable binding of RF1 and RF3 is mutually exclusive. On the other hand, a cryo-EM structure of ribosomes in complex with a deacylated tRNA in the P site, RF1, and RF3 in the apo form, that is, in the absence of added nucleotide, suggest that both factors can bind simultaneously to the ribosome (*Pallesen et al., 2013*). smFRET measurements carried out with post-release complexes in the presence of excess RF1 showed that the addition of RF3–GTP induced short-lived transitions from the L1-open to the L1-closed state which were not observed in the absence of RF3. This suggests that the two factors can bind to the ribosome simultaneously (*Sternberg et al., 2009*). No structural studies are available on the interaction of RF3 with RF2-bound complexes. The interaction of RF3 with the ribosomes prior to peptide release has not been studied.

Here, we use TIRF microscopy to monitor smFRET signals reporting on subunit rotation to follow changes in ribosome conformation in response to RF1, RF2 and RF3 and the binding of each individual release factor to the ribosome during termination. Our results demonstrate how the recruitment of release factors change the ribosome conformation in termination complexes, how the dissociation of the factors is achieved, show differences in the function of RF1 and RF2, and explain the importance of GTP binding and hydrolysis by RF3.

## Results

### RF1 and RF2 have distinct effects on ribosome dynamics

To monitor the rotation of the ribosomal subunits during termination, we utilized ribosomes with fluorescent labels attached to the small subunit protein S6 and the large subunit protein L9, S6-Cy5 and L9-Cy3, respectively. This FRET pair has been extensively characterized in both smFRET and ensemble kinetics experiments and reports on the formation of the non-rotated (N) or the rotated (R) state of the ribosome (*Cornish et al., 2008*; *Ermolenko et al., 2013*; *Sharma et al., 2016*). We prepared termination complexes on an mRNA which is translated up to the stop codon UAA recognized by both RF1 and RF2. The complexes contain a peptidyl-tRNA in the P site and have a stop codon in the A site; those complexes are denoted as pre-hydrolysis complexes (PreHC). In the absence of termination factors, PreHC is found predominantly in a state with the FRET efficiency of $0.73 \pm 0.01$ (denoted as 0.7 FRET state in the following) (*Figure 1A*, *Figure 1—figure supplement 1*; *Supplementary file 1*). Previous work has shown that this state corresponds to the N state of the ribosome (*Cornish et al., 2008*; *Qin et al., 2014*; *Sharma et al., 2016*). A small fraction of complexes shows a FRET state with an efficiency of $0.52 \pm 0.02$ (0.5 FRET state), which corresponds to the R state of the ribosome. While peptidyl-tRNA generally favors the N state, the ability of ribosomes with peptidyl-tRNA in the P site to adopt the R state at room temperature has been demonstrated previously by smFRET and cryo-EM (*Cornish et al., 2008*; *Fischer et al., 2010*; *Ling and Ermolenko, 2015*). The distribution of FRET efficiencies and thus the ratio between N and R conformations of PreHC is independent of the tRNA in the P site and of the presence of a single N-terminal amino acid (fMet) or a dipeptide (fMetPhe, fMetVal or fMetLys) at the P-site tRNA (*Figure 1—figure supplement 1*; *Supplementary file 1*). This finding has prompted us to use the PreHC with fMet-tRNA$^{\text{fMet}}$ in the P site and a stop codon in the A site as a minimal model system, following previous publications which used this approach to study termination (*Casy et al., 2018*; *Jin et al., 2010*; *Koutmou et al., 2014*; *Kuhlenkoetter et al., 2011*; *Pallesen et al., 2013*; *Pierson et al., 2016*; *Shi and Joseph, 2016*; *Sternberg et al., 2009*).

To probe the effect of RF1/RF2 binding on subunit rotation, we mixed PreHC with RF1 or RF2 at saturating concentrations of the factors (*Zavialov et al., 2002*). Peptide release was avoided by using RF1(GAQ) or RF2(GAQ) mutants which are catalytically deficient (*Frolova et al., 1999*; *Zavialov et al., 2002*) (*Figure 1—figure supplement 2A*). Binding of RF1(GAQ) to PreHC stabilizes the N state (*Figure 1B*). The fraction of the PreHC in the R state, albeit small, is somewhat higher with RF2(GAQ) than with RF1(GAQ) (*Figure 1B,C*).

The hydrolysis of the ester bond between the tRNA and the nascent peptide in PreHC leads to the formation of post-hydrolysis complex (PostHC). To prepare PostHC without the use of termination factors, we released nascent peptides with the help of puromycin, an analog of the A-site aminoacyl-tRNA that reacts with the peptidyl-tRNA in the P site to form peptidyl-puromycin (which then dissociates from the ribosome) and a deacylated tRNA in the P site. These complexes are denoted as PostHC*. The FRET histogram of PostHC* in the absence of the factors indicate the presence of two states, the 0.5 FRET (R) state and the 0.7 FRET (N) state (*Figure 1D*, *Supplementary file 1*). FRET time courses of individual ribosomes show reversible transitions between the N and R states (*Figure 1—figure supplement 1B*). The exact distribution of states depends on the P-site tRNA (*Figure 1—figure supplement 1*; *Supplementary file 1*) (*Cornish et al., 2008*) with tRNA$^{\text{fMet}}$ behaving similarly to tRNA$^{\text{Val}}$, thus underscoring the suitability of the minimal model system.

To test the effect of RF1 and RF2 on subunit rotation of PostHC, we added saturating concentrations of the wild type RF1 or RF2 to PreHC to allow peptide release. RF1 halts PostHC in the N state (*Figure 1E*, *Figure 1—figure supplement 2B,D*), in agreement with the previous smFRET study where RF1 binding stabilizes the L1 stalk in the open state (*Sternberg et al., 2009*). Binding of RF2 to PostHC shifts the equilibrium toward the N state, but not to the same extent as RF1 (*Figure 1F*, *Figure 1—figure supplement 2C,E*). Complexes with RF2 make occasional N to R transitions, in particular with RF2(GAQ) bound to PostHC (*Figure 1—figure supplement 2C*). These initial observations suggest that although both factors favor the N state, RF1 appears more efficient than RF2.

To further probe the potential differences between RF1 and RF2, we monitored subunit rotation in response to factor binding in real time. We injected catalytic amounts of release factors to PreHC and PostHC and recorded the time courses of FRET signal changes (*Figure 1—figure supplement 3*). RF1(GAQ) binding to PreHC does not change the FRET efficiency appreciably, as the complex is

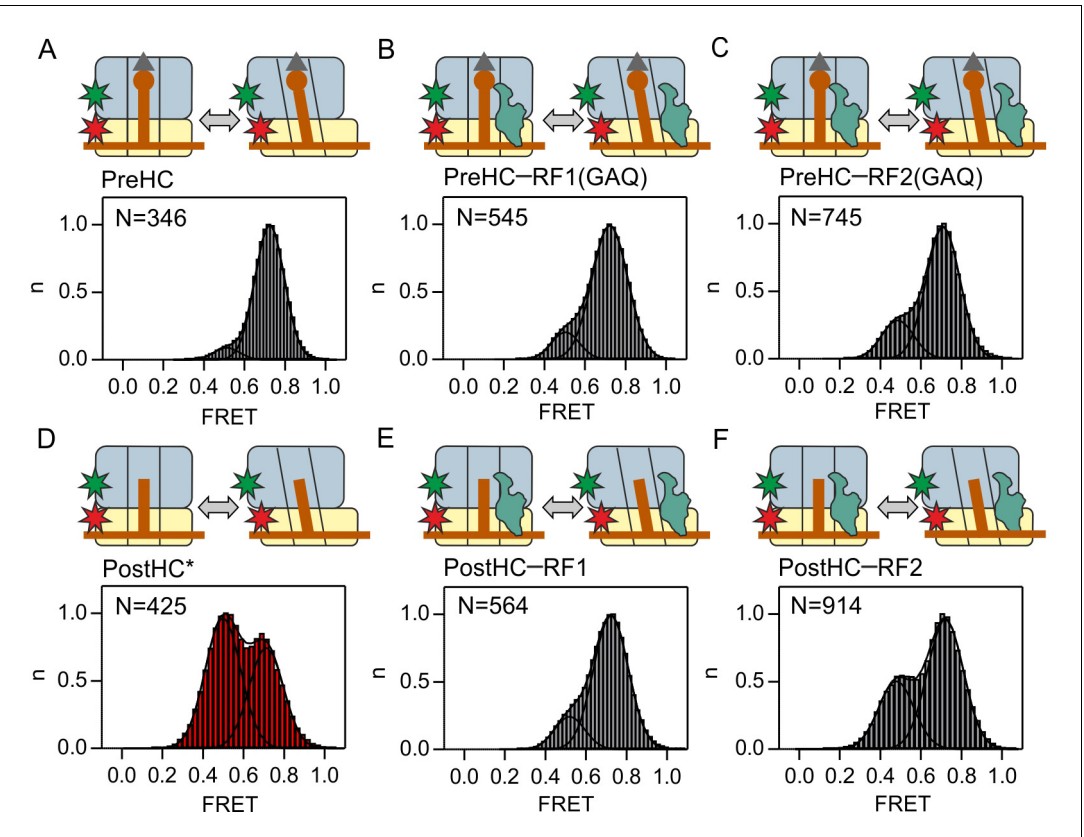

**Figure 1.** Subunit rotation of termination complexes in the presence of release factors. Histograms and Gaussian fits of normalized FRET distributions of S6/L9-labeled termination complexes in the presence of saturating RF1/RF2 concentrations. (**A,D**) PreHC and PostHC in the absence of RFs. In (**D**) PostHC* was generated by addition of puromycin to PreHC. (**B,E**) Same as A,D in the presence of RF1(GAQ) (1 μM) and RF1 (1 μM), respectively. PostHC is formed by the action of RF1. (**C,F**) Same as B,E in the presence of RF2(GAQ) (1 μM) and RF2 (1 μM), respectively. Cartoons show the complex composition. Grey triangles and brown circles represent the formyl group and the amino acid of fMet, respectively; stars indicate the positions of the Cy3 (green) and Cy5 (red) labels. The red shade of histogram in (**D**) indicates frequent reversible transitions between N and R states. The grey shade of all other histograms indicates that transitions were observed in less than 20% of traces. FRET values (**Supplementary file 1**) are calculated from three independent data sets. See also **Figure 1—figure supplement 1**, **Figure 1—figure supplement 2 Figure 1—figure supplement 3** and **Supplementary file 1**.

DOI: https://doi.org/10.7554/eLife.34252.003

The following figure supplements are available for figure 1:

**Figure supplement 1.** Subunit rotation of termination complexes in the absence of release factors.
DOI: https://doi.org/10.7554/eLife.34252.004

**Figure supplement 2.** Peptide hydrolysis by RF1/RF2(GAQ) mutants and subunit rotation of Post HC* monitored using FRET between S6-Cy5 and L9-Cy3.
DOI: https://doi.org/10.7554/eLife.34252.005

**Figure supplement 3.** Time-resolved subunit rotation.
DOI: https://doi.org/10.7554/eLife.34252.006

predominantly in the N state with or without the factor (**Figure 1—figure supplement 3A**). Also the binding of wild-type RF1 to PreHC with subsequent peptide release does not change the FRET efficiency (**Figure 1—figure supplement 3B**), supporting the notion that stabilization of the N state by RF1 is independent of peptide release (**Figure 1B,E**). PostHC without factor fluctuates between the N and R state; binding of RF1 to PostHC halts fluctuating ribosomes in the N state and prevents excursions to the R state (**Figure 1—figure supplement 3C**).

With RF2 the picture is somewhat different. PreHC–RF2(GAQ) is predominantly in the N state (**Figure 1—figure supplement 3D**). However, binding of wild type RF2 and peptide release shift the

distribution toward the R state (*Figure 1—figure supplement 3E*). The resulting PostHC fluctuates between N and R states as shown by synchronization of FRET traces to the first N to R transition. PostHC obtained by puromycin treatment also shows reversible N to R transitions which remain undisturbed by the addition of RF2 (*Figure 1—figure supplement 3F*). Although the binding of the factors is not directly monitored in these experiments, the differences in the rotation pattern suggest that RF1 and RF2 have distinct effects on ribosome dynamics. Such differences may result from a shorter residence time of RF2 compared to RF1 on the ribosome, which we tested in the following experiments.

## Binding of RF1 and RF2 to the ribosome

To measure how long the factors remain bound to the ribosome, we prepared Cy5-labeled RF1 and RF2, as well as the respective RF1/2(GAQ) mutants and ribosomes containing Cy3-labeled protein L11 (*Adio et al., 2015*; *Chen et al., 2011*; *Geggier et al., 2010*; *Holmberg and Noller, 1999*; *Stöffler et al., 1980*) (*Figure 2—figure supplement 1A*). The biochemical activity of labeled release factors was indistinguishable from that of the unlabeled counterparts (*Figure 2—figure supplement 1B,C*) and the factors were fully methylated (*Figure 2—figure supplement 2*). L11 constitutes part of the factor binding site (*Pallesen et al., 2013*; *Petry et al., 2005*; *Rawat et al., 2006*; *Rawat et al., 2003*). Recruitment of the factors to the ribosome is expected to result in high FRET efficiency. Binding of RF1 or RF1(GAQ) to either PreHC or PostHC results in a single FRET population centered at $0.72 \pm 0.02$ (0.7 FRET) (*Figure 2A–C*, *Supplementary file 1*). RF1 and RF1(GAQ) are stably bound to the ribosome, in agreement with previous biochemical reports on dissociation rates of RF1 and RF1(GAQ) ($0.005–0.1$ s$^{-1}$; [*Koutmou et al., 2014*; *Shi and Joseph, 2016*]). The estimated upper limit of the dissociation rate in our experiments is $0.2$ s$^{-1}$ (*Supplementary file 1*), defined by the photobleaching rate of the FRET dye pair with $k_{photobleaching} = 0.07–0.19$ s$^{-1}$ at the given imaging conditions (Materials and methods). Binding of RF2 or RF2(GAQ) to PreHC or PostHC leads to single FRET populations with efficiencies between 0.6 and 0.7 (*Figure 2D–F*, *Supplementary file 1*). However, the residence time of RF2 is much shorter compared to RF1 or RF1 (GAQ), with the $k_{off}$ values in the range from $0.8 \pm 0.1$ s$^{-1}$ to $1.3 \pm 0.2$ s$^{-1}$ (*Figure 2D–F*, *Supplementary file 1*). Peptide hydrolysis has no visible effect on RF1 and only a minor effect on RF2 dissociation (in the absence of RF3).

The difference in the dissociation rates of RF1 and RF2 implies that RF1 needs an auxiliary factor, RF3, to help it to dissociate from the ribosome, whereas RF2 may be able to recycle independent of RF3. This notion is consistent with previous reports (*Petropoulos et al., 2014*; *Zavialov et al., 2002*) and is further supported by our peptide hydrolysis turnover assay (*Figure 2—figure supplement 3A*). With catalytic amounts of RF1 in the absence of RF3, that is, when RF1 turnover depends on its intrinsic dissociation rate from the ribosome, termination is essentially blocked, whereas in the presence of RF3 RF1-mediated peptide release is very efficient. In contrast, even catalytic amounts of RF2 are sufficient to complete peptide release from PreHC, although RF3 accelerates the reaction by a factor of 10. Thus, RF3 is essential for RF1, but not for RF2 recycling. Addition of RF3 to PreHC–RF2(GAQ) or PostHC–RF2 complexes makes the complexes more dynamic (*Figure 2—figure supplement 3B–E*). Our results demonstrate that during canonical termination RF1 and RF2 interact with termination complexes in somewhat different ways, as they have different residence times on the ribosome and respond differently to the presence of RF3.

## Interaction with RF3–GTP

Next, we studied how RF3 affects ribosome dynamics and promotes the dissociation of RF1/RF2. To investigate the effect of RF3 on subunit rotation in the absence of RF1 or RF2, we added saturating concentrations of RF3 to S6/L9-labeled PreHC (*Figure 3A*, *Figure 2—figure supplement 1*). Binding of RF3 to PreHC, which in the absence of the factor is in the N state, strongly shifts the equilibrium toward the R state (*Figure 3A*), that is, RF3 has the opposite effect on subunit rotation than RF1 or RF2. The traces are now highly dynamic and show reversible N to R transitions (*Supplementary file 1*). This finding seems unexpected as ribosomes with peptidyl-tRNA in the P site favor the N state. However, previous cryo-EM and smFRET studies have indicated that those complexes can in fact adopt the R state (*Cornish et al., 2008*; *Fischer et al., 2010*; *Ling and Ermolenko, 2015*). Thus,

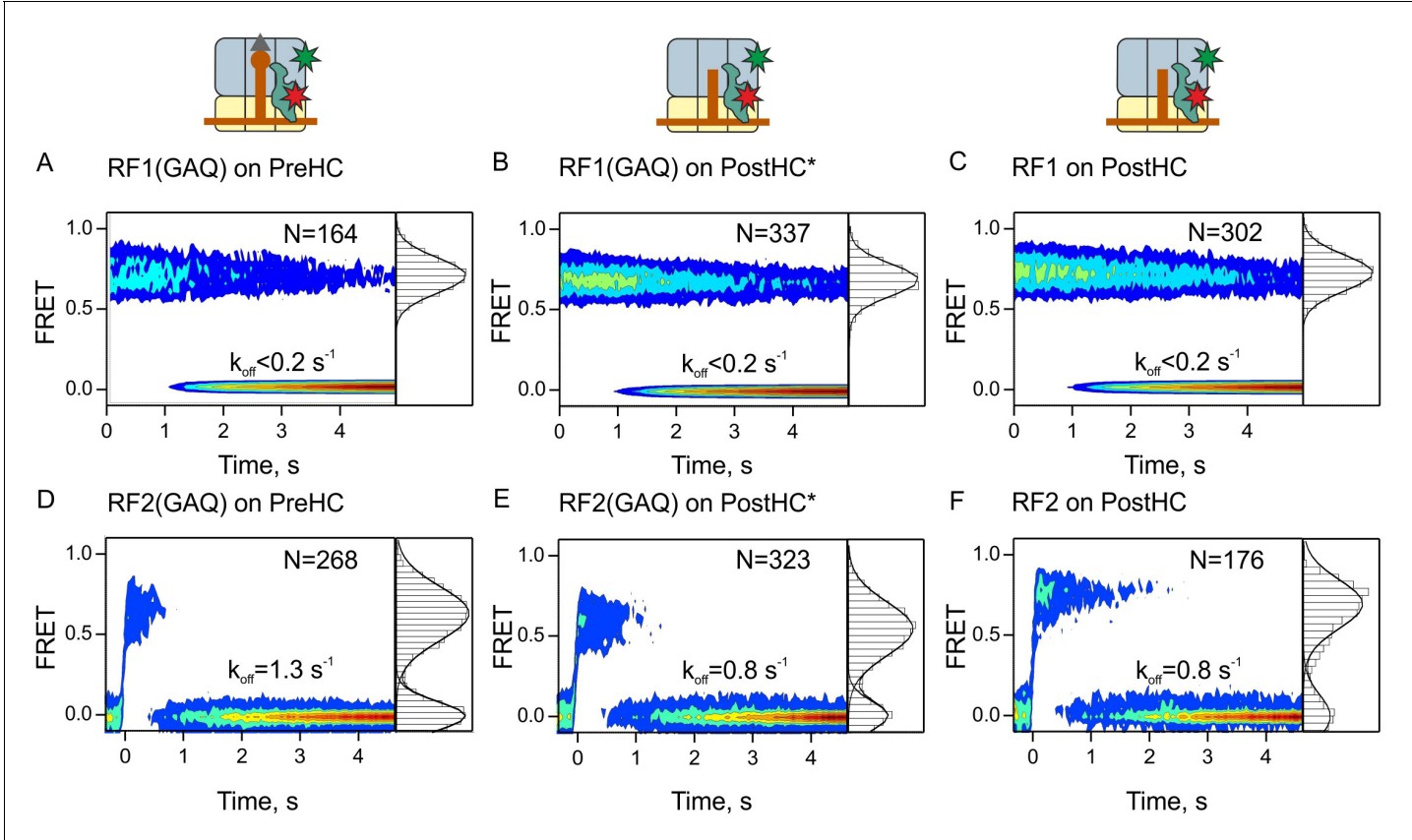

**Figure 2.** Residence times of RF1 and RF2 on PreHC and PostHC. (A–C) smFRET upon addition of RF1-Cy5 or RF1(GAQ)-Cy5 to PreHC or PostHC labeled at protein L11 with Cy3. FRET values (mean ± sd) center at $0.72 \pm 0.02$ (A), $0.71 \pm 0.01$ (B), and $0.71 \pm 0.01$ (C). (D–F) smFRET upon addition of RF2-Cy5 or RF2(GAQ)-Cy5 to PreHC or PostHC labeled at protein L11 with Cy3. FRET values (mean ± sd) center at $0.65 \pm 0.03$ (D), $0.56 \pm 0.05$ (E), and $0.70 \pm 0.04$ (F). Experiments were carried out with catalytic amounts of labeled release factors (10 nM). Individual traces were combined in contour plots. FRET histograms are plotted to the right of the contour plots. In (B,E) the peptide was released using puromycin. In the (C,F) peptide was released by RF1 and RF2, respectively. FRET signals were synchronized to the beginning of the FRET signal. $k_{off}$ is the rate of RF1 or RF2 dissociation. Mean FRET values and rate constants with sd were calculated from three independent data sets. See also *Figure 2—figure supplement 1*, *Figure 2—figure supplement 2*, *Figure 2—figure supplement 3* and *Supplementary file 1*.

DOI: https://doi.org/10.7554/eLife.34252.007

The following figure supplements are available for figure 2:

**Figure supplement 1.** Activity of the fluorescence-labeled RFs.

DOI: https://doi.org/10.7554/eLife.34252.008

**Figure supplement 2.** Quantification of release factor methylation by mass spectrometry.

DOI: https://doi.org/10.7554/eLife.34252.009

**Figure supplement 3.** Interplay between RF2 and RF3.

DOI: https://doi.org/10.7554/eLife.34252.010

RF3 appears to bias spontaneous fluctuations of peptidyl-tRNA in the PreHC, rather than induce a previously disallowed conformation.

To further characterize the conformational changes of the ribosome induced by RF3, we probed the position of the P-site tRNA relative to protein L1. We used a FRET pair with the donor label at the tRNA (fMet-tRNA^fMet-Cy3) and the acceptor label on ribosomal protein L1 (L1-Cy5). The two labels are close together and give a high FRET signal when ribosomes are in the L1-closed state and move apart to give a low FRET signal when ribosomes are in the L1-open state (*Fei et al., 2009*; *Fei et al., 2008*; *Munro et al., 2010a*; *Munro et al., 2010b*; *Munro et al., 2010c*; *Sternberg et al., 2009*). FRET histograms of PreHC in the absence of RF3 are dominated by a low-FRET population ($0.32 \pm 0.01$) and do not show transitions to other states (*Figure 3—figure supplement 1A*, *Supplementary file 1*). This indicates that ribosomes are predominantly in the L1-open state with

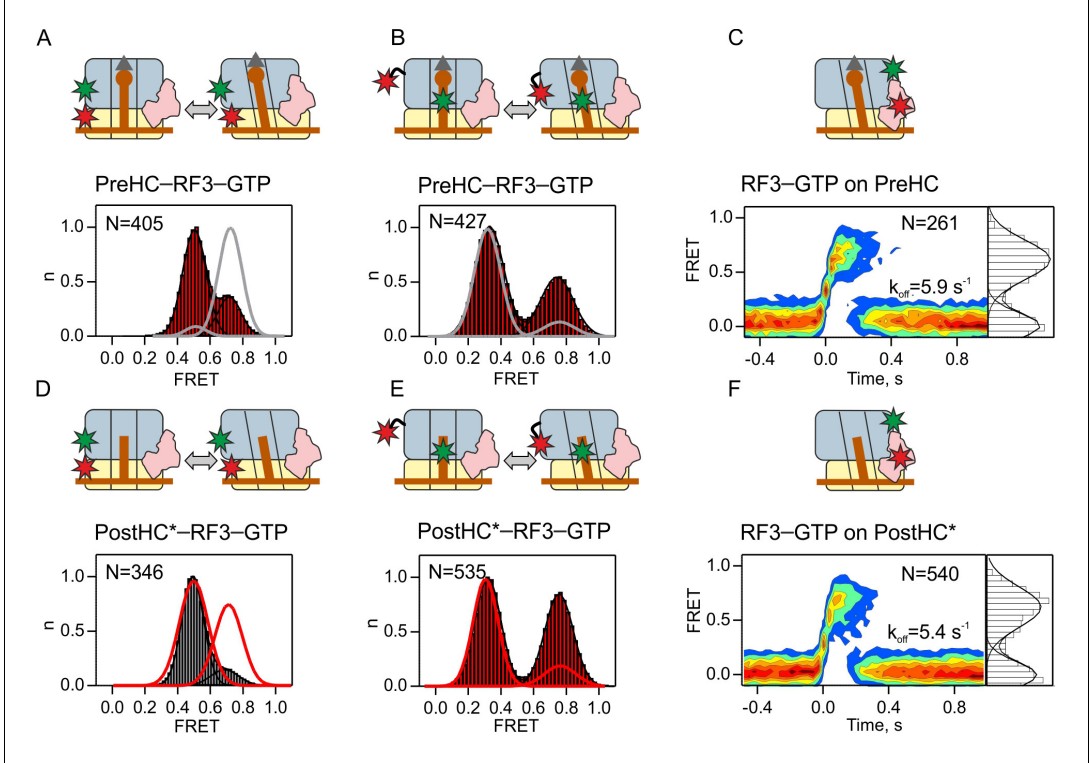

**Figure 3.** Interaction of RF3–GTP with termination complexes. (**A,D**) Subunit rotation of S6/L9-labeled PreHC and PostHC* in the presence of excess RF3 (1 µM) with GTP (1 mM). (**B,E**) Distribution of L1-open and L1-closed states in PreHC and PostHC* labeled at tRNA$^{fMet}$ and protein L1 in the presence of excess RF3 (1 µM) with GTP (1 mM). Smooth red and gray lines show distributions of states without RF3. Color code is the same as in histograms (red, frequent transitions between the states; gray, transitions in less than 20% of traces). (**C,F**) Contour plots representing the residence time of RF3-Cy5 (10 nM) on PreHC and PostHC* labeled at L11 by Cy3. FRET time courses were synchronized to the beginning of the first FRET event. FRET values (mean ±sd) 0.62 ± 0.02 (**C**) and 0.64 ± 0.04 (**F**) are from three independent data sets and plotted to the right of the contour plots. $k_{off}$ is the rate of RF3 dissociation. See also *Figure 3—figure supplement 1* and *Supplementary file 1*.

DOI: https://doi.org/10.7554/eLife.34252.011

The following figure supplement is available for figure 3:

**Figure supplement 1.** tRNA conformation in PreHC and PostHC*.

DOI: https://doi.org/10.7554/eLife.34252.012

fMet-tRNA$^{fMet}$ in the classic P/P state, in agreement with previous studies (*Cornish et al., 2009*; *Fei et al., 2008*; *Sternberg et al., 2009*). RF3 induces dynamic transitions from the low FRET state to a high FRET state (0.74 ± 0.02), which suggests that ribosomes transiently sample the L1-closed state with fMet-tRNA$^{fMet}$ in a hybrid-like P/E state (*Figure 3B*). This state is short lived ($k_{closed\rightarrow open}$ = 6.0 ± 0.8 s$^{-1}$; *Supplementary file 1*). The transition rate is faster than subunit rotation ($k_{R\rightarrow N}$ = 2.2 ± 0.4 s$^{-1}$) suggesting that the two processes are not tightly coupled, consistent with the previous smFRET work (*Munro et al., 2010a*; *Wasserman et al., 2016*) and cryo-EM reconstructions (*Fischer et al., 2010*).

We then monitored the dissociation of RF3 from PreHC using FRET between RF3–Cy5 and L11–Cy3 (*Figure 3C*). Labeling of RF3 did not change its catalytic properties (*Figure 2—figure supplement 1A,D*). The dissociation rate of RF3 from PreHC is $k_{off}$ = 5.9 ± 1.1 s$^{-1}$ (*Figure 3C*; *Supplementary file 1*). Thus, RF3-GTP can bind to PreHC and alter its conformation as shown by the rotation of subunits and movement of the peptidyl-tRNA into a P/E-like state, but the residence time of the factor on the ribosome is short.

To test whether the interaction of RF3 with termination complexes depends on peptide release, we then studied the effect of RF3 on subunit rotation of PostHC prepared by puromycin treatment (PostHC*) (*Figure 3D–F*). S6/L9-labeled PostHC* fluctuates between 0.5 and 0.7 FRET states (*Figure 1—figure supplement 1*). RF3 binding shifts the distribution toward the 0.5 FRET state,

indicating that the R state is stabilized (*Figure 3D*). The L1–tRNA FRET pair shows an enrichment of the high FRET state corresponding to the P/E state of the tRNA (*Figure 3E*). While subunits are stabilized in the R state and do not fluctuate toward N state, the L1-tRNA label shows reversible transitions between P/E and P/P conformations (*Supplementary file 1*). This suggests that also in PostHC subunit rotation and the formation of a hybrid-like state are not tightly coupled. Dissociation of RF3 from PostHC is as rapid as from PreHC, $k_{off}$ = 5.4 ± 1.3 s$^{-1}$ (*Figure 3F*, *Supplementary file 1*).

Our results show that RF3 facilitates the formation of the R state with the tRNA in a P/E-like orientation on both PreHC and PostHC. RF3 dissociation is not directly coupled to subunit rotation, as the rate of R to N transitions is lower than that of RF3 dissociation (*Figure 3*, *Supplementary file 1*). The residence time of RF3 on the ribosome is nearly identical on Pre- and PostHC which indicates that the presence of RF3 on the ribosome is not regulated by peptide release. We also note that the observed RF3 dissociation rates are much higher than the rate of GTP hydrolysis by RF3 (*Peske et al., 2014*; *Shi and Joseph, 2016*; *Zavialov et al., 2001*). This implies that rapid RF3 dissociation is independent of GTP hydrolysis.

## Interplay between RF1 and RF3

Next, we studied the interplay between RF1, RF3 and ribosomes during termination. We compared three different termination conditions including PreHC, PostHC* prepared by puromycin treatment, and PreHC which was converted to PostHC in situ upon the interaction with RF1. For each condition, we monitored (i) subunit rotation in the presence of saturating concentrations of both RF1 and RF3; (ii) RF1-Cy5 binding to the ribosome at saturating concentrations of unlabeled RF3; and (iii) RF3-Cy5 binding to the ribosome at saturating concentrations of unlabeled RF1 (*Figure 4*).

To follow the interactions of RF1 and RF3 with PreHC (*Figure 4A,B,C*), we again used the RF1 (GAQ) mutant, which ensures that peptidyl-tRNA in PreHC is not hydrolyzed. While RF1(GAQ) alone stabilizes the N state (grey line in *Figure 4A*; *Figure 1B*) and RF3 alone induces transitions from the N to the R state (*Figure 3A*), in the presence of saturating amounts of RF1(GAQ) and RF3 together the N and R states are almost equally populated (*Figure 4A*). Ribosomes show rapid reversible N to R transitions indicating that RF3 can promote subunit rotation even when RF1 is present. RF1(GAQ) binding to PreHC–RF3 results in a single FRET population centered at a FRET efficiency of 0.67 ± 0.02, similar to the 0.7 FRET when RF1 binds to the ribosome in the absence of RF3. The RF1 (GAQ) dissociation rate is low, <0.3 s$^{-1}$ (*Figure 4B,D*; *Supplementary file 1*), in agreement with previous reports (0.14 ± 0.02 s$^{-1}$, [*Koutmou et al., 2014*]). RF3 binds to PreHC–RF1 (*Figure 4C*), but the FRET efficiency for the RF3-L11 pair is reduced compared to the complex in the absence of RF1 (0.51 ± 0.03 and 0.62 ± 0.02 in the presence and absence of RF1, respectively; *Supplementary file 1*). Thus, the orientation of RF3 on the PreHC is shifted by RF1, whereas the position of RF1 appears unchanged, at least with respect to L11. The rate of RF3 dissociation in the presence of RF1(GAQ) is 1.3 ± 0.1 s$^{-1}$, which is higher than the dissociation rate of RF1(GAQ), but about fivefold slower than that of RF3 in the absence of RF1 (*Figure 3C*; *Figure 4D*; *Supplementary file 1*), indicating that RF1 stabilizes the binding of RF3 to PreHC. Dwell time distributions for N or R state in the presence of RF1(GAQ) and RF3 are biphasic, suggesting the presence of two populations of each complex. The majority of ribosomes display rapid transitions (>70%, $k_{N \to R}$ = 5.9 ± 0.6 s$^{-1}$, $k_{R \to N}$ = 2.9 ± 0.4 s$^{-1}$; *Figure 4D*; *Supplementary file 1*) that are faster than RF1 or RF3 dissociation, indicating that subunits can rotate while both factors are bound to the ribosome. Low rotation rates ($k_{N \to R}$ = 1.30 ± 0.07 s$^{-1}$, $k_{R \to N}$ = 0.80 ± 0.05 s$^{-1}$; <30% of ribosomes) are also observed with RF3 alone and thus may represent subunit rotation after RF1 dissociation (*Supplementary file 1*). The observed shift of PreHC–RF1 from the predominantly N to a fluctuating ensemble of N and R states upon RF3 addition, together with the altered RF3 position and the decreased RF3 dissociation rate when the two factors are bound suggest that the complex undergoes conformational adjustments when RF1 and RF3–GTP are bound simultaneously.

Next, we monitored subunit rotation in PostHC. For better comparison with the results obtained with PreHC and RF1(GAQ), we first prepared PostHC* by puromycin treatment of PreHC and studied the interactions with RF1(GAQ) and RF3–GTP (*Figure 4E,F,G,H*). In the presence of RF1 and RF3, the majority of complexes undergo rapid N to R transitions and the equilibrium is shifted toward the R state (*Figure 4E*). The mean FRET efficiency for RF1(GAQ) binding to PostHC*–RF3 changes to 0.67 ± 0.02 compared to 0.50 ± 0.03 for RF1(GAQ) binding to PreHC–RF3 (*Figure 4B,F*) or 0.71 ± 0.01 for binding to complexes in the absence of RF3 (*Figure 2A–C*). The decrease in FRET

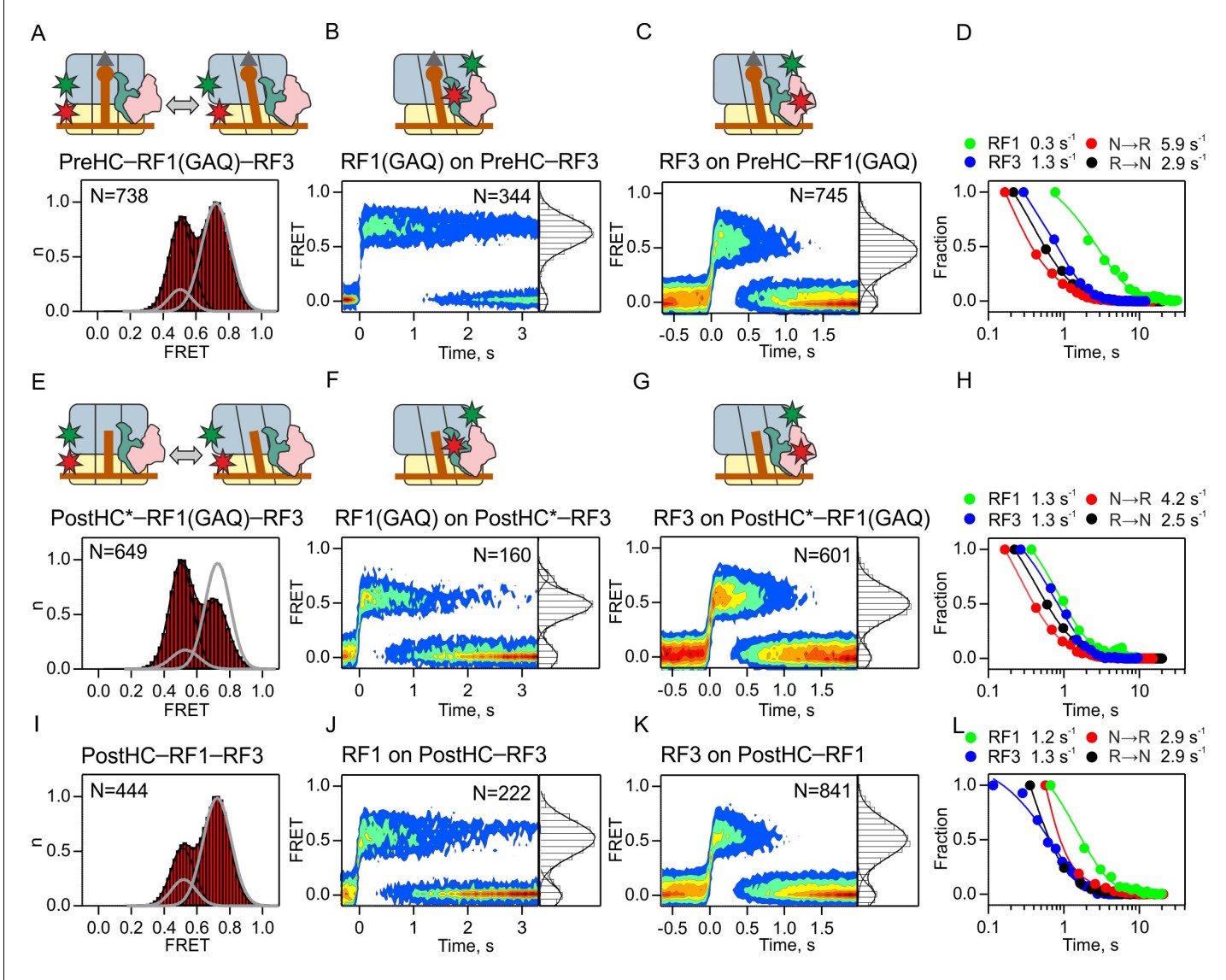

**Figure 4.** Interplay between RF1 and RF3–GTP. (A,E,I) Subunit rotation of S6/L9-labeled Pre- and PostHC measured at saturating RF1 and RF3–GTP concentrations (1 µM each). Grey line represents FRET distribution in the absence of RF3. (B,F,J) Contour plots representing the residence time of RF1-Cy5/RF1(GAQ)-Cy5 ribosomes labeled at protein L11 by Cy3 in the presence of excess RF3 (1 µM). Time courses were synchronized to the beginning of the FRET signal. FRET values (mean ±sd) are 0.67 ± 0.02 (B), 0.50 ± 0.03 and 0.76 ± 0.02 (F), and 0.53 ± 0.04 (I). (C,G,K) Contour plots representing the residence time of RF3-Cy3 on ribosomes labeled at protein L11 by Cy3 in the presence of excess RF1 or RF1(GAQ) (1 µM). FRET values (mean ± sd) are 0.51 ± 0.03 (C), 0.51 ± 0.03 (G), and 0.51 ± 0.03 (K). (D,H,L) Comparison of the rates of RF1 and RF3 dissociation and subunit rotation. (A–D) Interactions with PreHC. (E–H) Interactions with PostHC* obtained by puromycin treatment. (I–L) Interactions with PostHC which is formed in situ using RF1. All values are mean ± sd from three independent data sets. See also *Supplementary file 1*.

DOI: https://doi.org/10.7554/eLife.34252.013

efficiency suggests that peptide release allows a rearrangement of the complex which alters the position of RF1 relative to L11. The FRET efficiency for RF3 binding to either PreHC–RF1(GAQ) or PostHC*–RF1(GAQ) is 0.51 ± 0.03 (*Figure 4C,G*), as compared to 0.62 and 0.64, respectively, for binding to PreHC or PostHC in the absence of RF1 (*Figure 3C,F*). This suggests that the position of RF3 on PreHC and PostHC is affected by the presence of RF1, but not by peptide release (*Figures 3F* and *4G*). The dissociation rates are 1.3 ± 0.2 s$^{-1}$ and 1.3 ± 0.1 s$^{-1}$ for RF1(GAQ) and RF3, respectively (*Figure 4F–H*; *Supplementary file 1*). A small fraction (8%) of complexes that release RF1(GAQ) slowly ($k_{off}$ = 0.12 ± 0.07 s$^{-1}$) is likely due to incomplete peptide hydrolysis by

puromycin. The rotation rates ($k_{N \to R}$ = 4.20 ± 0.08 s$^{-1}$, $k_{R \to N}$ = 2.50 ± 0.03 s$^{-1}$) are somewhat higher than RF1 and RF3 dissociation rates, but the most prominent effect of peptide release is the acceleration of RF1 dissociation from <0.3 s$^{-1}$ to 1.3 ± 0.2 s$^{-1}$ (*Figure 4D,H*; *Supplementary file 1*).

Similar effects are observed when instead of puromycin we used wild-type RF1 to convert PreHC to PostHC (*Figure 4I–L*): at saturating concentrations of RF1 and RF3 the R state of PostHC is enriched and the complexes show reversible N to R transitions (*Figure 4I*; *Supplementary file 1*). RF1 and RF3 are bound to PostHC in the 0.5 FRET state (*Figure 4J,K*; *Supplementary file 1*). The dissociation rates are 1.2 ± 0.4 s$^{-1}$ for RF1 (>70% of ribosomes; *Figure 4J*; *Supplementary file 1*) and 1.3 ± 0.2 s$^{-1}$ for RF3 (*Figure 4K*; *Supplementary file 1*). Thus, RF1 stabilizes the binding of RF3 on PreHC or PostHC, whereas RF3 destabilizes RF1 binding, but only after peptide release. Peptide release also allows an adjustment in the positions of both factors relative to L11. Thus, peptide release is a major determinant for RF1, but not RF3, dissociation.

Because the kinetics of subunit rotation is faster than RF3 and RF1 dissociation, it remains unclear from which state, N or R, the factors dissociate. To test whether R state formation is required for RF3 dissociation, we used the antimicrobial peptide apidaecin 137 (Api) as a tool to trap RF1 on termination complexes. Api binds into the exit tunnel of PostHC and prevents RF1/RF2 dissociation (*Florin et al., 2017*). When we monitor subunit rotation in the presence of saturating concentrations of RF1, RF3 and Api, the PostHC–RF1–RF3–Api complex is stalled in the N state (*Figure 5A*). In the absence of RF1 Api does not alter the relative fraction of N and R states induced by RF3 (*Figure 5B*). In the PostHC–RF1–Api–RF3 complex, RF1 is stably bound in the 0.7 FRET state (*Figure 5C*). RF3 is bound in 0.5 FRET state and dissociates with the rate of 1.2 ± 0.1 s$^{-1}$ (*Figure 5D*). These data suggest that RF3 can dissociate independent of subunit rotation from termination complexes that are exclusively in the N state as well as from termination complexes that show mixed N and R populations.

## The role of GTP binding and hydrolysis

By analogy with other GTPases, GTP hydrolysis by RF3 is expected to regulate the dissociation of RF3 from the ribosome. In contrast to all other GTPases, RF3 was suggested to bind to the PostHC-RF1 complex in the GDP-bound form; the ribosome-induced rapid release of GDP should stabilize RF3 binding, while subsequent GTP binding induces a conformational change of the ribosome and the release of RF1 (*Sternberg et al., 2009*; *Zavialov et al., 2002*). We first tested these models using a biochemical turnover peptidyl-tRNA hydrolysis assay and compared the effect of different nucleotides on factor recycling (*Figure 6A,B*). When both RF1 and RF3 are sub-stoichiometric to PreHC, such that 10 cycles of RF1 and RF3 turnover are required to convert all PreHC to PostHC, peptide release is only observed in the presence of GTP (*Figure 6A*). In excess of RF3, when only RF1 has to turnover, efficient peptide release is observed with wild type RF3 in the presence of GTP, GTPγS or GDPNP (*Figure 6B*). Also RF3(H92A)–GTP, a RF3 mutant deficient in GTP hydrolysis, induces efficient recycling of RF1, contrary to previous reports (*Gao et al., 2007*), but consistent with a recent kinetic study (*Shi and Joseph, 2016*). Apo-RF3 has no activity, again consistent with previous reports (*Shi and Joseph, 2016*; *Zavialov et al., 2002*). The low activity in the presence of GDP is most likely due to a minor contamination with GTP. Thus, GTP hydrolysis is not required for RF1 recycling but is necessary to ensure recycling of RF3, while the apo and GDP forms of RF3 appear inactive.

Next, we sought to understand how different nucleotides affect the interaction of RF3 with termination complexes. RF3–GTP promotes R state formation, which can be used as readout for the ribosome interaction with RF3 in complex with different nucleotides (*Figure 6—figure supplement 1*). PreHC in the presence of excess RF3–GDP or RF3 in the apo form are predominantly in the N state and do not show transitions to the R state (*Figure 6—figure supplement 1A,B*); the ratio of N and R states is identical to that in the PreHC in the absence of RF3 (*Figure 1A*). Also RF1-bound PostHC in the presence of excess RF3–GDP or apo-RF3 are predominantly in the N state and the distribution of states is very similar to that in RF1-bound termination complexes (*Figure 6—figure supplement 1C,D* and *Figure 1E*, respectively). Together, these experiments suggest that RF3-GDP and apo-RF3 are not able to induce the R state in termination complexes. By analogy, smFRET experiments monitoring the position of the L1 stalk show that addition of RF3-GDP or apo-RF3 does not change the ribosome conformation (*Sternberg et al., 2009*). For a more direct observation of RF3-GDP or apo-RF3 binding to the ribosome, we made an attempt to follow FRET between RF3-Cy5 and termination

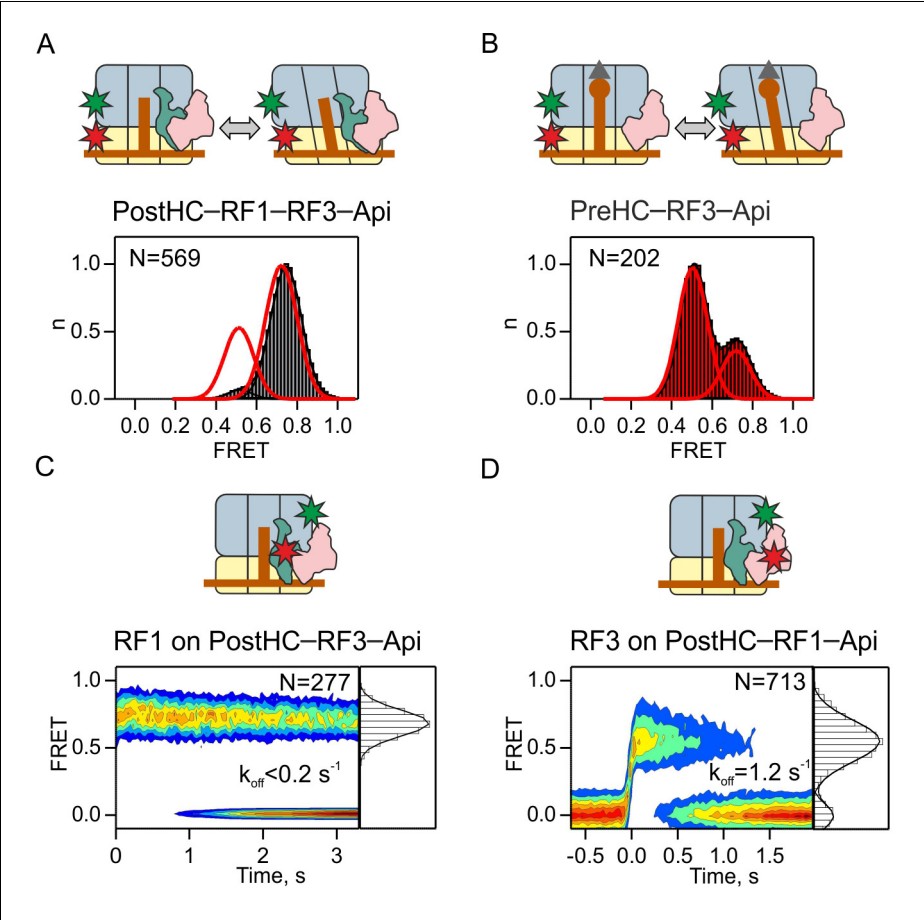

**Figure 5.** Dissociation of RF3 from RF1-bound PostHC in the presence of Api. (**A,B**) Subunit rotation of S6/L9-labeled Pre- and PostHC with or without RF1 (1 µM), with saturating RF3 concentrations (1 µM) and Api (1 µM). Red lines in (**A**) and (**B**) represent FRET distribution in the absence of Api. (**C**) Contour plot representing the residence time of RF1-Cy5 on PostHC-Cy3 in the presence of saturating RF3 concentration (1 µM) and Api (1 µM). FRET values (mean ± sd) center at 0.71 ± 0.01. (**D**) Contour plot representing the residence time of RF3-Cy5 on PostHC-Cy3 in the presence of saturating RF1 concentration (1 µM) and Api (1 µM). FRET values (mean ± sd) center at 0.55 ± 0.04. All values are mean ± sd from three independent data sets. See also ***Supplementary file 1***.
DOI: https://doi.org/10.7554/eLife.34252.014

complexes labeled at protein L11 with Cy3. However, we did not find any FRET events indicative of RF3 binding in the presence of GDP or with apo-RF3 (data not shown). These observations suggest that although RF3-GDP or apo-RF3 must bind to PostHC–RF1 in some way, because this interaction accelerates nucleotide exchange in RF3 (***Koutmou et al., 2014***; ***Peske et al., 2014***; ***Shi and Joseph, 2016***; ***Zavilov et al., 2001***) the interaction must be transient and does not engage the factor at its binding site at L11 unless GTP is bound.

We then asked whether GTP hydrolysis by RF3 is required to induce subunit rotation. We replaced GTP with a non-hydrolysable analog, GDPNP, which is extensively used in structural studies. RF3–GDPNP can bind to the PreHC or PostHC obtained by addition of RF1 and induces formation of the R state, albeit not to the same extent as RF3–GTP and with fewer transitions between N and R states (***Figure 6C,D***). The same tendencies are observed with RF3(H92A)–GTP or RF3–GTPγS (***Figure 6—figure supplement 1E,F,G***). The exact fraction of the R state and dynamic ribosomes depends on the choice of nucleotide, which may indicate that the ability of RF3–GDPNP or RF3–GTPγS to form a stable complex with the ribosome is reduced compared to RF3–GTP.

We then tested whether GTP hydrolysis is required for RF3 dissociation from the ribosome. The dissociation rate of RF3–GDPNP from PostHC* in the absence of RF1 is $k_{off} = 0.34 \pm 0.04$ s$^{-1}$, much

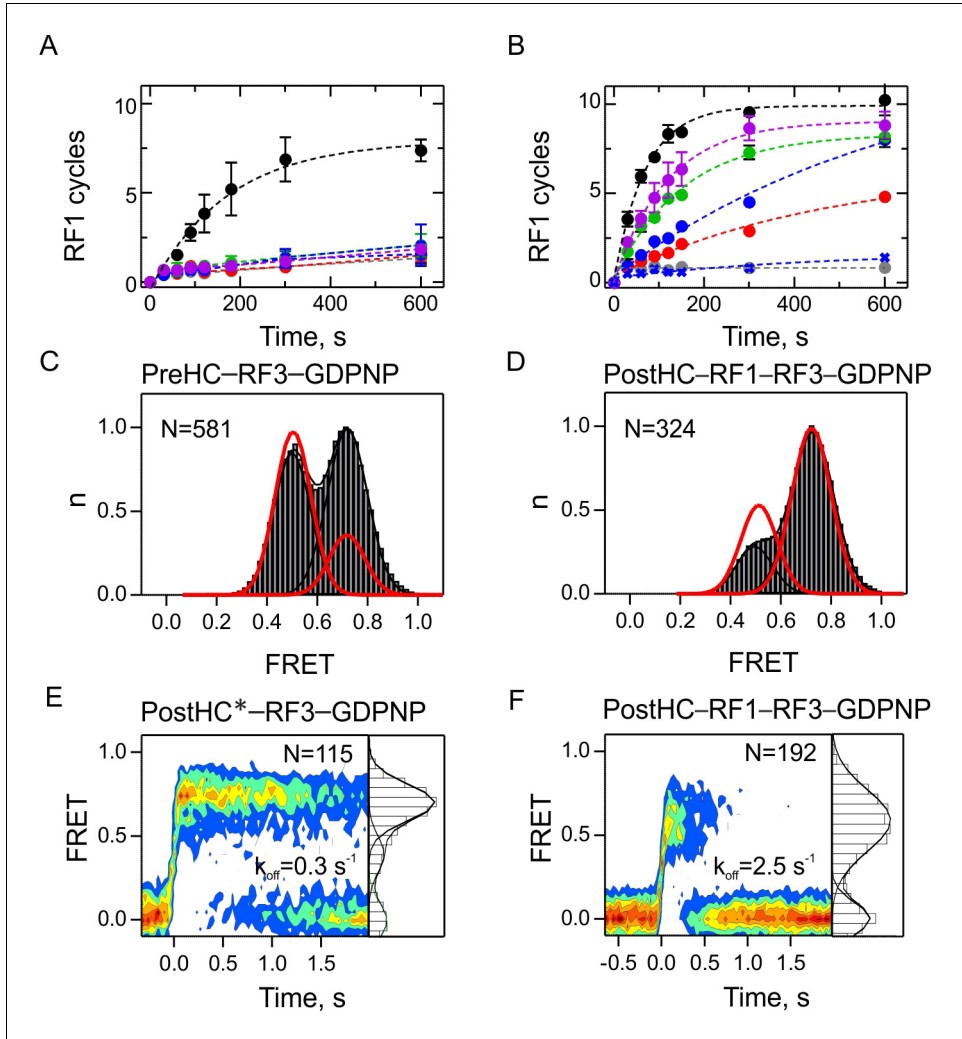

**Figure 6.** The role of GTP hydrolysis for RF1 and RF3 recycling. (A,B) Effect of different nucleotides on peptidyl-tRNA hydrolysis (GTP, black circles; GTPγS, green circles; GDPNP, blue circles; GDP, red circles; no nucleotide, grey circles) or in the presence of RF3(H92A) and GTP (purple circles). Control experiments are in the absence of RF3 (blue crosses). Error bars represent the range of two technical replicates. (A) Peptide hydrolysis was performed by incubating PreHC (100 nM) with RF3 (10 nM) and the respective nucleotides (1 mM); reactions were started with the addition of RF1 (10 nM). (B) Same as in (A), but at 1 μM RF3. (C,D) FRET distribution reporting on subunit rotation of S6/L9-labeled PreHC in the presence of saturating amounts of RF3–GDPNP (C) or RF1 with RF3–GDPNP (D) (1 μM RF each). Red lines represent the distribution of FRET states with RF3–GTP. (E,F) Contour plots representing the residence time of RF3-Cy5 (10 nM) on PostHC labeled at L11 by Cy3 in the presence of GDPNP (1 mM) without RF1 (E) or (F) in the presence of saturating RF1 concentration (1 μM). FRET values (mean ± sd) center at 0.71 ± 0.01 and 0.40 ± 0.01 (E) and 0.58 ± 0.03 (F). All values are mean ± sd from three independent data sets. See also *Figure 6—figure supplement 1*, *Figure 6—figure supplement 2* and *Supplementary file 1*.
DOI: https://doi.org/10.7554/eLife.34252.015

The following figure supplements are available for figure 6:

**Figure supplement 1.** Effect of the nucleotide bound to RF3 on subunit rotation.
DOI: https://doi.org/10.7554/eLife.34252.016

**Figure supplement 2.** Dissociation of RF1 from PostHC mediated by RF3 in the absence of GTP hydrolysis.
DOI: https://doi.org/10.7554/eLife.34252.017

lower than with GTP (*Figure 6E* and *Supplementary file 1*). In contrast, in the presence of saturating RF1 concentrations dissociation of RF3–GDPNP from PostHC-RF1 is as rapid as with GTP (*Figure 6F* and *Supplementary file 1*), indicating that GTP hydrolysis is not essential when RF1 is present. Experiments with RF3–GTPγS gave very similar results (*Figure 6—figure supplement 2*). At saturating RF3 concentrations, dissociation of RF1 from PostHC is independent of GTP hydrolysis (*Figure 6—figure supplement 2*), consistent with the biochemical data (*Figure 6B*). In the simplest model, these findings can be interpreted as an indication for the role of GTP hydrolysis in RF3 dissociation from termination complexes in the absence of RF1. They also explain why RF1 turnover is impaired at sub-stoichiometric RF3 concentrations when GTP hydrolysis is blocked (*Figure 6A*): those RF3 molecules that bind to ribosomes lacking RF1 remain stalled if GTP is not hydrolyzed, thereby depleting the pool of RF3 which has to turnover to stimulate RF1 dissociation. Thus, the only reaction where GTP hydrolysis or an authentic GTP conformation appears to play an essential role is the dissociation of RF3 from termination complexes in the absence of RF1.

## Discussion

Our experiments show how release factors navigate through the landscape of possible ribosome conformations during translation termination (*Figure 7A*). Release factors not only change the ratio between the N and R states, but also alter the fraction of the ribosomes that make transient fluctuations between the states. Each factor alone has its distinct signature on ribosome conformation and dynamics. Binding of RF1 to either PreHC or PostHC favors the static N state; protein L1 adopts an open conformation, which correlates with a classical state of the P-site tRNA. The N state of the ribosome–RF1 complex has been also captured by structural studies (*James et al., 2016*; *Korostelev et al., 2008*; *Laurberg et al., 2008*; *Petry et al., 2005*; *Weixlbaumer et al., 2008*). Surprisingly, we find that PreHC–RF2 is more dynamic, and has a higher fraction of the R states than the complex with RF1. Furthermore, RF2 can dissociate equally well from the PreHC and PostHC and is

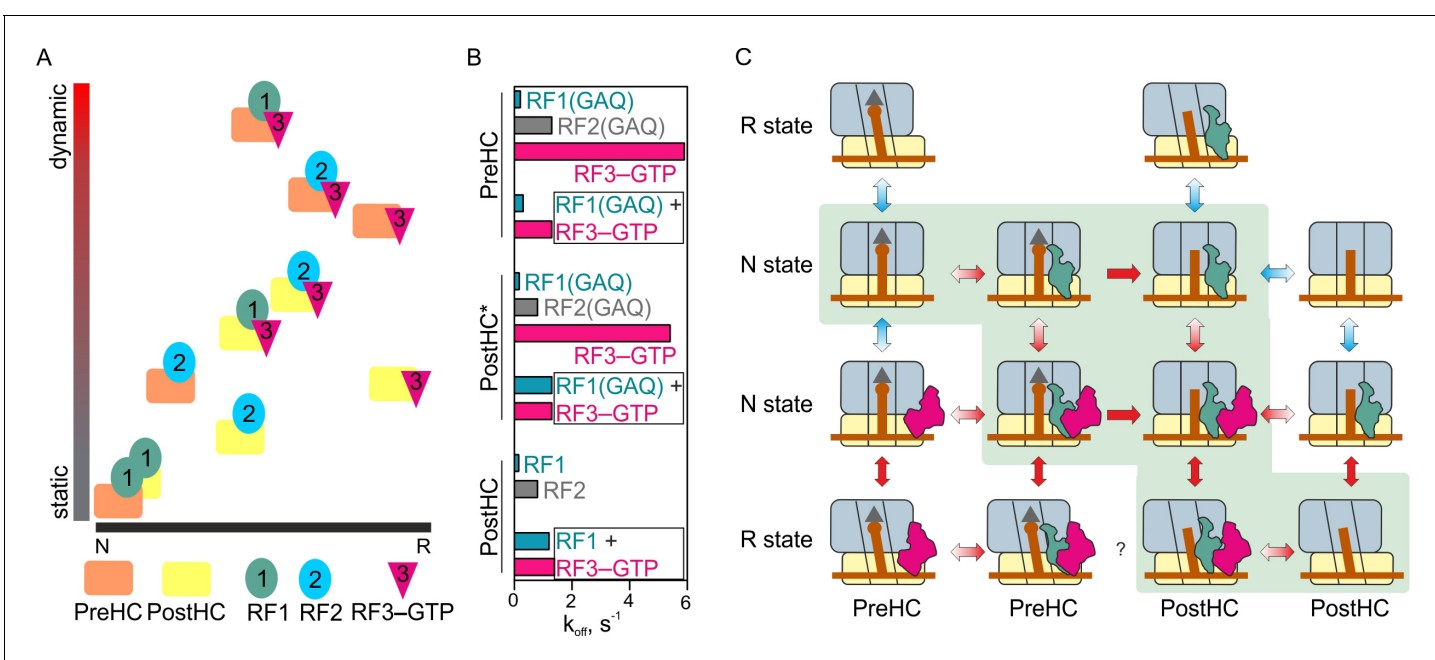

**Figure 7.** The mechanism of translation termination. (**A**) Ribosome dynamics in the presence of RF1, RF2, and RF3. Ribosome fluctuations are color-coded from static (gray) to dynamic (red) and correlated to the fraction of N and R state in the respective complex. (**B**) Summary of the dissociation rate constants of RF1, RF2 and RF3 from different type of complexes. Bars representing the dissociation of RF1 are colored teal, RF2 gray, RF3 magenta. (**C**) The landscape of ribosome conformations with RF1 and RF3. The ribosome states (N and R, PreHC and PostHC) are indicated. Red arrows indicate rapid reaction, blue arrows static or slowly exchanging states, with the preferential direction indicated by color gradient; single-headed arrows indicate irreversible steps of peptidyl-tRNA hydrolysis. See also *Supplementary file 1*.
DOI: https://doi.org/10.7554/eLife.34252.018

less dependent on the action of RF3 than RF1 (*Figure 2—figure supplement 3A*, *Figure 7B*). With its high dissociation rate, RF2 action may depend on the ratio between the rate of peptide release and factor dissociation, for example, if the rate of peptidyl-tRNA hydrolysis is about 10 s$^{-1}$ (*Indrisiunaite et al., 2015*; *Kuhlenkoetter et al., 2011*) and the rate of RF2 dissociation is ~1 s$^{-1}$ (this paper), the factor can achieve efficient peptide release before dissociating. Thus, RF1 and RF2 – albeit fulfilling a similar function during canonical termination – differ in their ability to affect ribosome dynamics.

Binding of RF3–GTP to termination complexes shifts the conformational distribution toward the R state (*Figure 7A*). The PreHC–RF3 complex is dynamic, whereas the PostHC–RF3 is stabilized in the R state, consistent with the previous smFRET work (*Sternberg et al., 2009*) and structural studies (*Gao et al., 2007*; *Jin et al., 2011*; *Zhou et al., 2012*). After peptide release, RF1 and RF3 or RF2 and RF3 together shift the distribution of ribosome conformations towards the middle of the dynamic spectrum (*Figure 7A*). The rates of ribosome fluctuations are in the range of 2–6 s$^{-1}$, somewhat faster than in the absence of the factors, 0.5–2.6 s$^{-1}$ (*Supplementary file 1*).

One open question is what drives the dissociation of RF1 and RF3 from the ribosome (*Figure 7B*). Dissociation of RF1 from the static N state is very slow. RF3 accelerates the dissociation, which correlates with increased ribosome dynamics and frequent transitions from N to R state. However, dynamic transitions alone are not sufficient to induce RF1 dissociation from the ribosome, because peptide release is crucial to allow RF1 to dissociate rapidly. Peptide release leads to a change in the orientation of RF1 with respect to L11. On the other hand, peptide release alone is not sufficient, as the dissociation rate of RF1 from the PostHC is slow in the absence of RF3. Thus, RF1 dissociation is promoted by the concerted action of RF3, which stimulates subunit rotation and may directly displace RF1 from its original binding site, and by peptide release, which allows a conformational adjustment of RF1.

RF3 dissociation is independent of peptide release or the ribosome dynamics, but is affected by the presence of RF1 or RF2, which stabilize RF3 binding to the ribosome and change conformation/ position of RF3 relative to L11. In the presence of RF1, RF3 efficiently dissociates from the N state even in the absence of GTP hydrolysis (this paper and [*Shi and Joseph, 2016*]). The order of RF1 and RF3 dissociation appears random, because the rates of factor release are quite similar and the exact sequence depends on experimental conditions (this paper; [*Koutmou et al., 2014*; *Shi and Joseph, 2016*]). In those cases where RF1 happens to dissociate before RF3 has left the ribosome, GTP hydrolysis completes RF3 recycling. In summary, subunit rotation, peptide release, conformational changes of the factors, and GTP hydrolysis together drive dissociation of RF1 and RF3. However, kinetically these movements are not directly coupled, that is the dissociation rates of the factors and the rates of subunit rotation are independent of each other but are individually defined by the dynamic properties of the complex. Thus, translation termination is a stochastic process that utilizes loosely coupled motions of its players to complete protein synthesis and release the newly synthesized nascent chain toward its cellular destination.

Our results lead to the following model of translation termination for RF1 (*Figure 7C*). Among all possible reaction routes, two appear most likely, either via RF1 binding to PreHC, followed by peptide release and RF3–GTP recruitment, or through simultaneous binding of RF1 and RF3–GTP to PreHC followed by peptide release. The resulting complex PostHC–RF1–RF3–GTP can make rapid transitions between the N and R states. RF1 and RF3 change their relative positions and can now both rapidly dissociate from the ribosome. The order of events is not deterministic: multiple ribosome conformations, ribosome dynamics and the lack of strong coupling between the rates of subunit rotation and the dissociation of RF1 and RF3 seem characteristic features of RF1-dependent termination.

This work provides an unexpected view on the role of nucleotide exchange and GTP hydrolysis by RF3. Although RF3-GDP or apo-RF3 can bind to the ribosome carrying RF1/RF2 (*Peske et al., 2014*; *Zavialov et al., 2001*), this interaction does not result in the recruitment of the factor to its binding site at the vicinity of L11. In vitro in the absence of GTP, apo-RF3 can form a relatively stable complex with PostHC–RF1 (*Pallesen et al., 2013*; *Shi and Joseph, 2016*), but this binding does not alter the dynamics of subunit rotation and does not accelerate RF1 dissociation (this paper and [*Sternberg et al., 2009*]). Rather, the GTP-bound form of RF3 is required to stimulate ribosome dynamics and RF1 dissociation from PostHC. Given the moderate difference in the affinities of RF3 for GTP and GDP, at cellular concentrations a large fraction of RF3 is in the GTP form. Furthermore,

given the high GTP association rate, apo-RF3 will be immediately converted into the functionally active GTP form (*Peske et al., 2014*); thus, the apo-RF3–ribosome complex can only be a transient intermediate. The present experiments, most of which are performed in the presence of a GTP regeneration system, which does not allow for accumulation of the GDP- or apo-form of RF3, show efficient factor binding, peptide release and factor recycling. We thus have no indication for an active role of nucleotide exchange or for an essential role of the GDP- or the apo-form of RF3 in termination at cellular conditions and we consider the respective models unlikely.

Unexpectedly, our data suggest that GTP hydrolysis or an authentic GTP-bound form of RF3 are required to release RF3 that is arrested on the ribosome in the absence of RF1. At the first glance, the low dissociation rate of RF3–GDPNP from the ribosome appears to contradict the results of the experiments with RF3–GTP, which show that factor dissociation is not coupled to GTP hydrolysis (*Figure 3C,F*). We hypothesize that upon binding to the ribosome, RF3 can either form an initial binding complex from which the factor can dissociate rapidly, or enter an engaged complex, from which RF3 can only dissociate after GTP hydrolysis (*Figure 6E*). In principle, this should result in biphasic dissociation time courses of RF3-GTP with a second slow phase corresponding to the rate of GTP hydrolysis, which we did not observe. However, in the presence of GTP the fraction of RF3 molecules that enter the engaged state may be too small to capture. As RF3–GDPNP appears to have a lower affinity to the ribosome than RF3–GTP, the transient initial RF3–ribosome complex might be too short-lived to be detected and only the stable engaged complexes are captured. Alternatively, GDPNP, as well as GTPγS or the RF3(H92A) mutant may induce a conformation that hinders RF3 from dissociation but is hardly populated in the presence of GTP; in this case, the effects are purely conformational and not due to GTP hydrolysis as such.

Available structures of ribosome-bound RF3 suggest that RF3 is arrested on ribosomes in the R state (*Gao et al., 2007*; *Jin et al., 2011*; *Zhou et al., 2012*). This could explain why PostHC, with its higher propensity to be in the R state than the PreHC, is more efficient in stimulating GTP hydrolysis by RF3 (*Zavialov et al., 2002*). In this respect, RF3 appears to be an unusual GTPase that differs from other translational GTPases, such as EF-G, EF-Tu and IF2, where GTP hydrolysis is coupled to key steps on the reaction pathway of the factors and is required on all ribosome complexes. Rather, the internal clock of the RF3 GTPase (*Peske et al., 2014*) acts as a rescue mechanism to release RF3 recruited to complexes that do not contain RF1. This scenario is realistic at the concentrations of factors in the cell where RF3 is much more abundant than RF1 (*Schmidt et al., 2016*).

The smFRET data presented here for a simple model system present a starting point to study dynamics of more natural termination complexes containing long peptide nascent chains. While model termination systems are fully functional in all steps of termination and the rate of GTP hydrolysis by RF3 is similar with the fM-Stop and fMFTI-Stop termination contexts (*Zavialov and Ehrenberg, 2003*), the length of the nascent peptide and the nature of the P-site tRNA may attenuate the ribosome dynamics. While currently such complexes are biochemically too heterogeneous to study, further development of smFRET techniques toward multicolor detection and better time resolution may provide a tool to decipher the dynamics of these heterogeneous assemblies.

# Materials and methods

## Key resources table

| Reagent type (species) or resource | Designation | Source or reference | Identifiers | Additional information |
|---|---|---|---|---|
| Strain, strain background (*E. coli*) | JW3947-1 | Keio collection | CGSC#: 12041 | E. coli *rplA* knockout strain |
| Sequence-based reagent | Start-stop mRNA | IBA (Göttingen) | N/A | RNA oligonucleotide: 5'-GGCAAGGAGGUAAAUAAU GUAAACGAUU-3' |
| Sequence-based reagent | mMetStop | IBA (Göttingen) | N/A | RNA oligonucleotide: 5'-Biotin-CAACCUAAAACUUACACA CCCGGCAAGGAGGUAAAUAAU GUAAACGAUU-3' |

*Continued on next page*

*Continued*

| Reagent type (species) or resource | Designation | Source or reference | Identifiers | Additional information |
|---|---|---|---|---|
| Sequence-based reagent | mMetPheStop | IBA (Göttingen) | N/A | RNA oligonucleotide: 5'-Biotin-CAACCUAAAACUUACACACCC GGCAAGGAGGUAAAUAAUGUUU UAAACGAUU-3 ' |
| Sequence-based reagent | mMetLysStop | IBA (Göttingen) | N/A | RNA oligonucleotide: 5'-Biotin-CAACCUAAAACUU ACACACCCGGCAAGGAGGUA AAUAAUGAAGUAAACGAUU-3 ' |
| Sequence-based reagent | mMetValStop | IBA (Göttingen) | N/A | RNA oligonucleotide: 5'-Biotin-CAACCUAAAACUUAC ACACCCGGCAAGGAGGUAAAU AAUGGUUUAAACGAUU-3 ' |
| Peptide, recombinant protein | RF2(GAQ) (recombinant protein) | PMID: 12419223 | | |
| Peptide, recombinant protein | RF1(GAQ) (recombinant protein) | PMID: 12419223 | | |
| Peptide, recombinant protein | RF1(S167C) (recombinant protein) | PMID: 19597483 | | Single-cysteine RF1 |
| Peptide, recombinant protein | RF2(C273) (recombinant protein) | This paper | | Single-cysteine RF2 |
| Peptide, recombinant protein | RF3(L233C) (recombinant protein) | This paper | | Single-cysteine RF3 |
| Peptide, recombinant protein | L1(T202C) (recombinant protein) | PMID: 18471980 | | Single-cysteine L1 |
| Peptide, recombinant protein | Apidaecin137 (API) (peptide) | NovoPro Biosciences Inc. | N/A | |
| Chemical compound, drug | Cy3-maleimide | GE Healthcare | PA23031 | |
| Chemical compound, drug | Cy5-maleimide | GE Healthcare | PA25031 | |
| Software, algorithm | Matlab | MathWorks | | |
| Software, algorithm | Prism GraphPad | GraphPad Software, La Jolla California USA, www.graphpad.com | | |
| Software, algorithm | Matlab code vbFRET | http://vbfret.source forge.net/ | | Described in *Bronson et al. (2009)* |

## Buffers

All smFRET experiments were performed in imaging buffer (50 mM Tris-HCl pH 7.5, 70 mM $NH_4Cl$, 30 mM KCl, 15 mM $MgCl_2$, 1 mM spermidine, 8 mM putrescine, 2.5 mM protocatechuic acid, 50 nM protocatechuate-3,4-dioxygenase (from *Pseudomonas*), 1 mM Trolox (6-hydroxy-2,5,7,8-tetramethyl-chromane-2-carboxylic acid), and 1 mM methylviologen). Peptide hydrolysis experiments were performed in $TAKM_7$ buffer (50 mM Tris-HCl pH 7.5, 70 mM $NH_4Cl$, 30 mM KCl, 7 mM $MgCl_2$).

## Labeled ribosomes, release factors and tRNA

The preparation and functional characterization of ribosomes labeled with Cy3 at protein L11 and double-labeled at S6-Cy5 and L9-Cy3 was carried out as described (*Adio et al., 2015*; *Sharma et al., 2016*). *E. coli* strain lacking L1 were obtained from the Keio collection (CGSC#: 12041) and ΔL1 ribosomes purified according to the protocol used for native ribosomes (*Rodnina and Wintermeyer, 1995*). A single cysteine was introduced at position T202 of L1 and the protein purified as described in *Fei et al. (2008)*. L1(T202C) was fluorescence labeled with Cy5-maleimide (GE Healthcare) and purified using a 5 ml HiTrap SP HP cation exchange chromatography column (GE Healthcare). ΔL1 ribosomes were reconstituted by incubation with a 5-fold molar excess

of L1-Cy5 for 30 min at 37°C. Excess protein was removed by centrifugation through a 30% sucrose cushion in 50 mM Tris-HCl pH 7.5, 70 mM NH$_4$Cl, 30 mM KCl, 15 mM MgCl$_2$, pH 7.5.

The RF2 construct was cloned from the *E. coli* K12 strain and contains the natural T246A replacement (*Wilson et al., 2000*). Catalytically impaired RF1(G234A) (RF1(GAQ)) and RF2(G251A) (RF2 (GAQ)), and the respective single-cysteine variant RF1(S167C) (*Sternberg et al., 2009*; *Wilson et al., 2000*), RF2(C273) and RF3(L233C) were generated by Quickchange mutagenesis according to the standard protocol. Native cysteines were replaced by serine or alanine based on the sequence conservation analysis performed using the Consurf database. RF1 and RF2 were purified and in vitro methylated as described (*Kuhlenkoetter et al., 2011*). RF3 was purified by affinity chromatography on a Ni-IDA column (Macherey-Nagel) followed by ion exchange chromatography on a HiTrapQ column (*Peske et al., 2014*). Prior to labeling, methylated RF1 and RF2 were incubated for 30 min with a 10-fold molar excess of TCEP (Sigma) at room temperature (RT). Cy5 maleimide (GE Healthcare) was dissolved in DMSO and added to the proteins (5- to 10-fold molar excess). Labeling was performed for 2 hr at RT and quenched by addition of a 10-fold molar excess of 2-mercaptoethanol over dye. Excess dye was removed by gel filtration on a PD-10 column (GE Healthcare). tRNA$^{fMet}$ was labeled at position s$^4$U8 with Cy3-maleimide (*Fei et al., 2010*) and aminoacylated and purified as described (*Milon et al., 2007*).

## mRNA

All mRNAs used in the smFRET experiments are labeled with biotin at the 5′end and were purchased from IBA (Göttingen, Germany). The following sequences were used: mMetStop
   5′-Biotin-CAACCUAAAACUUACACACCCGGCAAGGAGGUAAAUAAUGUAAACGAUU-3′
mMetPheStop
   5′-Biotin-CAACCUAAAACUUACACACCCGGCAAGGAGGUAAAUAAUGUUUUAAACGAUU-3′
mMetLysStop
   5′-Biotin-CAACCUAAAACUUACACACCCGGCAAGGAGGUAAAUAAUGAAGUAAACGAUU-3′
mMetValStop
   5′-Biotin-CAACCUAAAACUUACACACCCGGCAAGGAGGUAAAUAAUGGUUUAAACGAUU-3 ′

For the peptide hydrolysis experiments, ribosome complexes were assembled on the synthetic model mRNA, 5′-GGCAAGGAGGUAAAUAAUGUAAACGAUU-3′ (IBA) with a start codon followed by a stop codon.

## Sample preparation for smFRET TIRF experiments

Initiation complex formation was carried out by incubating ribosomes (100 nM) with a three-fold excess of IF1, 2 and 3, fMet-tRNA$^{fMet}$, mRNA and 1 mM GTP in TAKM$_7$ for 30 min at 37°C. To form initiation complexes with fMet-tRNA$^{fMet}$-Cy3, equal amounts of ribosomes and tRNA were used. In case of the mRNA coding for fMetStop, the initiation complex was used as PreHC. To generate PreHC on other mRNAs, an equal volume of ternary complex was added containing EF-Tu (1 μM) incubated with GTP (1 mM), phosphoenolpyruvate (3 mM) and pyruvate kinase (0.1 mg/ml) in TAKM$_7$ for 15 min at 37°C, followed by addition of Phe-tRNA$^{Phe}$, Lys-tRNA$^{Lys}$ or Val-tRNA$^{Val}$ (500 nM). Addition of EF-G (100 nM) and GTP (1 mM) induced tRNA translocation to form PreHC that contains peptidyl tRNA in the P site and displays the UAA stop codon in the A site.

## TIRF experiments

Complexes were diluted to 1 nM with smFRET buffer (50 mM Tris-HCl, 70 mM NH$_4$Cl, 30 mM KCl, 15 mM MgCl$_2$, 1 mM spermidine and 8 mM putrescine). Biotin/PEG functionalized cover slips were incubated for 5 min at room temperature with the same buffer containing additionally BSA (10 mg/ml) and neutravidin (1 μM) (Thermo Scientific). Excess neutravidin was removed by washing the cover slip with buffer containing BSA (1 mg/ml). Ribosome complexes were applied to the surface and immobilized through the mRNA-biotin:neutravidin interaction. Images were recorded at a rate of 30 frames/s after exchanging the buffer with imaging buffer at room temperature (22°C) (*Adio et al., 2015*).

To monitor subunit rotation of L9/S6-labeled ribosomes in the presence of release factors at steady-state conditions, imaging buffer was supplemented with RF1, RF2 and/or RF3 (1 μM each). In experiments with RF1(GAQ) or RF2(GAQ), the observation time was limited to <10 min in order to

minimize peptide hydrolysis due to residual factor activity. In experiments monitoring subunit rotation by RF3 in the GTP form or in complex with non-hydrolysable GTP analogs, imaging buffer was additionally supplemented with the energy recycling system (1 mM GTP or 1 mM GDPNP or 1 mM GTPγS, 3 mM phosphoenolpyruvate and 0.1 mg/ml pyruvate kinase). FRET signals reporting on the time course of subunit rotation during termination were obtained by injecting RF1 or RF2 (100 nM) in imaging buffer to immobilized PreHC or PostHC.

To measure FRET signals reporting on the residence time of labeled release factors on PreHC or PostHC labeled at protein L11 with Cy3, the complexes were immobilized on the cover slip. Movies were recorded upon addition of Cy5-labeled RF1, RF2 or RF3 to a final concentration of 10 nM in imaging buffer. To study the residence time of Cy5-labeled RF3 or to study the residence time of Cy5-labeled RF1 or RF1(GAQ) on ribosomes in the presence of unlabeled RF3, imaging buffer was supplemented with unlabeled RF3 (1 µM), GTP (1 mM), phosphoenolpyruvate (3 mM) and pyruvate kinase (0.1 mg/ml). To study the residence time of Cy5-labeled RF3 in the presence of RF1, imaging buffer was supplemented with unlabeled RF1 (1 µM), GTP (1 mM), phosphoenolpyruvate (3 mM) and pyruvate kinase (0.1 mg/ml).

To monitor FRET signals reporting on the conformation of the P-site tRNA PreHC or PostHC labeled on protein L1(C202-Cy5) and on fMet-tRNA$^{fMet}$(thioU8-Cy3) or tRNA$^{fMet}$(U8-Cy3) were immobilized on the coverslip. Movies were recorded upon addition of imaging buffer or imaging buffer containing RF3 (1 µM). In experiments with RF3 imaging, buffer was additionally supplemented with the energy recycling system (1 mM GTP or 1 mM GDPNP or 1 mM GTPγS, 3 mM phosphoenolpyruvate and 0.1 mg/ml pyruvate kinase) (*Sternberg et al., 2009*).

## Data analysis

Fluorescence time courses for donor (Cy3) and acceptor (Cy5) were extracted as described (*Adio et al., 2015*; *Fei et al., 2008*; *Roy et al., 2008*). A semi-automated algorithm (Matlab) was used to select anti-correlated fluorescence traces (correlation coefficient <0.1) exhibiting characteristic single fluorophore fluorescence intensities (*Adio et al., 2015*). Time traces for further analysis were selected from the dataset by choosing only those traces that contained single photobleaching steps for Cy3 and Cy5 (as recommended in [*Fei et al., 2008*]). The bleed-through of the Cy3 signal into the Cy5 channel was corrected using an experimentally determined coefficient (~0.13 in our experimental system [*Adio et al., 2015*]). All trajectories were smoothed over three data points. FRET efficiency was defined as the ratio of the measured emission intensities, Cy5/(Cy3 +Cy5) (*Roy et al., 2008*). FRET-histograms were fitted to Gaussian distributions using Matlab code (*Adio et al., 2015*). Mean FRET values (mean ±sd) and population distribution (p=area under the curve ± sd) were calculated from three independent datasets and are summarized in *Supplementary file 1*.

The vbFRET software package (http://vbfret.sourceforge.net/) (*Bronson et al., 2009*) was used for hidden Markov model (HMM) analysis of the FRET data. Time trajectories with only one transition per trace and with the FRET changes of less than 0.1 were excluded from further kinetic analysis (*Fei et al., 2008*; *Sternberg et al., 2009*). Individual time-resolved FRET traces were compiled into FRET probability density plots (contour plots) (*Blanchard et al., 2004*; *Munro et al., 2007*). For the experiments measuring subunit rotation of PostHC upon binding of RF1 in real time, FRET traces were synchronized at the transition to the stable N state. For the experiments measuring subunit rotation of PreHC upon binding of RF2 in real time, FRET traces were synchronized to the first N to R transition. In experiments measuring the residence time of labeled release factors, FRET traces are synchronized to the beginning of the FRET event reporting on the binding of the factor to the ribosome. One-dimensional histograms at the right side of the contour plots summarize FRET values of the first 10–30 time frames (0.3–1.0 s) of the FRET signals. The photobleaching rates of the S6/L9-FRET pair were estimated as described (*Adio et al., 2015*) from the non-fluctuating 0.7 FRET trajectories obtained with PreHC, PreHC-RF1(GAQ) and PostHC-RF1, as well as from the non-fluctuating 0.5 FRET trajectories of PostHC*-RF3(GTP); the photobleaching rates were in the range of 0.07–0.19 s$^{-1}$, comparable to 0.05–0.3 s$^{-1}$ in (*Sternberg et al., 2009*). Dwell times of individual FRET states in traces with multiple FRET states were calculated from idealized traces (*Bronson et al., 2009*). Dwell time histograms were fitted to either one- or two-exponential function. Rates (k) were calculated by taking the inverse of dwell times. Rate constants ± standard deviations were determined from three

independent datasets as described in *Fei et al. (2011)*; *Sternberg et al. (2009)*; *Wasserman et al. (2016)* and summarized in *Supplementary file 1*.

## Peptide hydrolysis assay

PreHC was prepared as described (*Peske et al., 2014*) and purified through sucrose cushion centrifugation. After centrifugation, ribosome pellets were resuspended in TAKM$_7$, frozen in liquid nitrogen and stored at $-80°C$. The extent of initiation was better than 95% as determined by nitrocellulose filtration and radioactive counting. PreHC (100 nM) was incubated with RF3 at the indicated concentration and nucleotide (1 mM) for 15 min at 37°C. Pyruvate kinase (0.1 mg/ml) and phosphoenol pyruvate (3 mM) were added in all experiments performed in the presence of GTP. Time courses were started by addition of RF1 or RF2 (10 nM). Samples were quenched with a solution containing TCA (10%) and ethanol (50%). After centrifugation (30 min, 16,000 g), the amount of released f[$^3$H]Met in the supernatant was quantified by radioactive counting.

## Acknowledgements

We thank Marija Liutkute for preparation of RF2-Cy5 and Olaf Geintzer, Franziska Hummel, Sandra Kappler, Christina Kothe, Anna Pfeifer, Theresia Uhlendorf, Tanja Wiles, and Michael Zimmermann for expert technical assistance. We thank Dr. H Urlaub and the bioanalytical mass spectrometry facility for assistance with the analysis of RF1/RF2 methylation. The work was supported by the grants of the Deutsche Forschungsgemeinschaft (SFB860 for MVR and SA).

## Additional information

### Funding

| Funder | Grant reference number | Author |
| --- | --- | --- |
| Deutsche Forschungsge-meinschaft | SFB 860 | Sarah Adio Marina V Rodnina |
| Max-Planck-Institute for Bio-physical Chemistry | Open-access funding | Marina V Rodnina |

The funders had no role in study design, data collection and interpretation, or the decision to submit the work for publication.

### Author contributions

Sarah Adio, Conceptualization, Formal analysis, Supervision, Validation, Investigation, Methodology, Writing—original draft, Writing—review and editing; Heena Sharma, Prajwal Karki, Wolf Holtkamp, Resources, Investigation, Methodology, Writing—review and editing; Tamara Senyushkina, Software, Formal analysis, Validation, Investigation, Methodology, Writing—review and editing; Cristina Maracci, Conceptualization, Supervision, Investigation, Methodology, Writing—review and editing; Ingo Wohlgemuth, Investigation, Methodology, Writing—review and editing; Frank Peske, Conceptualization, Supervision, Writing—review and editing; Marina V Rodnina, Conceptualization, Supervision, Funding acquisition, Writing—original draft, Writing—review and editing

### Author ORCIDs

Prajwal Karki (iD) http://orcid.org/0000-0001-6187-6506
Cristina Maracci (iD) http://orcid.org/0000-0003-3810-7180
Marina V Rodnina (iD) http://orcid.org/0000-0003-0105-3879

### Decision letter and Author response

Decision letter https://doi.org/10.7554/eLife.34252.022
Author response https://doi.org/10.7554/eLife.34252.023

## Additional files

### Supplementary files

• Supplementary file 1. Related to *Figures 1–7*. Quantitative analysis of conformational dynamics during termination.
DOI: https://doi.org/10.7554/eLife.34252.019

• Transparent reporting form
DOI: https://doi.org/10.7554/eLife.34252.020

### Data availability

All data generated or analysed during this study are included in the manuscript and supporting files.

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
