## [Decision Letter]

[Editors’ note: a previous version of this study was rejected after peer review, but the authors submitted for reconsideration. The first decision letter after peer review is shown below.]

Thank you for submitting your work entitled "Dynamics of ribosomes and release factors during translation termination in *E. coli*" for consideration by *eLife*. Your article has been evaluated by a Senior Editor and three reviewers, one of whom is a member of our Board of Reviewing Editors.

We have received comments from three experts in the field and these reviewers have discussed their independent views to reach a decision. Based on these discussions and the individual reviews below, we regret to inform you that your work will not be considered further for publication in *eLife*.

While all three reviewers found your data on the function of bacterial termination factors RFs1-3 interesting, and all appreciated the major effort to develop the story, the consensus view was that: (1) there were substantive experimental issues that still need to be addressed and (2) even if these concerns are addressed, the manuscript will lack a cohesive narrative that would be of broad interest to the readers of *eLife*.

The reviews are included but the critical points that the reviewers focused on were:

a) All three reviewers were surprised to see arguments suggesting that the ribosome can assume a rotated conformation independent of tRNAs moving into a hybrid state of binding (though this surprising observation was never very clearly articulated and rationalized to the reader). One possibility would be to form complexes with more authentic peptidyl-tRNAs (rather than fMet-tRNA) since it is very difficult to imagine that these peptidyl-tRNAs could migrate to the E site. Alternatively, the rotated state being observed with an intermediate FRET value (~0.5) may not be the true rotated state but instead a different intermediate or binding state (controls need to be performed to confirm the authenticity of this FRET state as rotated).

b) The differences between RF2 and RF1 behavior were very interesting but given the reduced affinity of RF2 relative to RF1, there are a number of experimental concerns. Is RF2 fully methylated? Is the site of modification (a different site than used for RF1) problematic (data not provided)? Are complexes fully saturated or would incomplete binding explain some of the heterogeneity (i.e. it is compositional)? This latter possibility seems critical to establish since RF2 does seem to bind more weakly. The biphasic nature of many of the dwell time histograms are suggestive of such compositional heterogeneity.

c) Finally, the data are not presented with error bars or statistical analyses of significance.

*Reviewer #1:*

This manuscript by Adio et al. uses smFRET, ensemble FRET, and ensemble kinetic assays to probe conformational changes on the ribosome associated with peptide release and release factor binding (RF1, RF2, and RF3). Several new findings are described that refute the older model of RF3 function, particularly the effect of nucleotide binding and hydrolysis, though that particular point has been addressed by the Rodnina lab previously. The data for RF2 appear to contradict previous structural work though the experiments here were performed on tRNAs lacking a true peptide in the exit tunnel. Finally, the fact that ribosomal subunit rotation and factor binding and dissociation are not kinetically coupled in a straightforward manner makes it difficult to propose a clear mechanistic model.

Conclusions from this work:

1) While RF1 stabilizes the non-rotated state, RF2 enriches the rotated state of the ribosome and dissociates quickly (a new and controversial finding). RF2 dissociates quickly from either pre or post termination complexes even without RF3.

2) RF3-GTP stabilizes the rotated state (previously known) and dissociates quickly from pre or post termination complexes.

3) Binding of RF3 to ribosomes bound with RF1 changes the conformation of both factors and stabilizes RF3 binding. RF1 and RF3 dissociation appears random; if RF3 dissociates first (from the non-rotated state), it does not require GTP hydrolysis, but if RF1 dissociates first, RF3 requires GTP hydrolysis to leave from the rotated state. (This finding is new and important).

4) Rotation, peptide release, GTP hydrolysis are not kinetically coupled in a straightforward manner.

Concerns:

In the first paragraph of the subsection “RF1 and RF2 have distinct effects on ribosome dynamics”, the authors state that the tRNA in the PreHC does not affect FRET dynamics (and for these experiments they use fMet-tRNA). However, the figures seem to depict a dipeptidyl-tRNA. This should be clarified in the text, figures, and figure captions. More importantly, it raises the question as to why they observe the rotated (R) state in the PreHC upon either RF2 or RF3 binding. Is this allowed rotation specific to the use of this minimal substrate fMet-tRNA? What would happen with a longer peptidyl-tRNA that extends into the exit tunnel? This is an important question because this finding suggests (contrary to the literature) that the hybrid tRNA states and rotated state of the ribosome can be separated.

In Figure 3 and Figure 6G, release of RF3 is much faster than GTP hydrolysis, but one of the main conclusions is that RF3 requires GTP hydrolysis to be released from the rotated state (Figure 6D, H).

*Reviewer #2:*

This paper is an attempt to use single-molecule FRET to dissect the events during termination of bacterial translation. Termination in bacteria involves class I release factors RF1 and RF2 that recognize the stop codon in the A site of the ribosome and trigger peptide release. A second GTPase class II release factor RF3 binds to the ribosome and accelerates dissociation of release factors from the ribosome, setting the stage for recycling (the final stage of translation).

Although the actions of RF1 and RF2 have become clearer over the years, the action of RF3 remains less so. In particular, there have been competing models. A model proposed by Ehrenberg's lab suggested that the ribosome functions as a GTP exchange factor for RF3 that is closely related to its mechanism.

This paper expands on recent work by both the Rodnina and Green labs to show a number of things. Firstly, it suggests that RF1 and RF2 do not function in precisely the same way. RF2 is less stably bound and does not require the action of RF3 to dissociate. Moreover, while RF1 strongly favors the non-rotated state without RF3, the same is not true for RF2. Building on previous papers in the Rodnina/Wintermeyer and Green labs, the paper provides strong evidence that RF3 in fact binds the ribosome in the GTP form, and that GTP hydrolysis is not a prerequisite for RF1 or RF2 release. Moreover, the apo and GDP forms are not capable of catalyzing release of RF1.

Finally, the paper shows that the temporal order of dissociation of RF1 and RF3 is random. Rather the complex favors a form that facilitates dissociation of both.

This is a nice study that is a clear advance on previous papers and helps clarify some of the confusion in this area. The difference between RF1 and RF2 is surprising.

The implication of their control (with just tRNAs) is that the rotated state with RF1,2/3 is similar that reached with tRNAs during translocation. However, a P/E tRNA state is not possible for the preHC complex, because it requires a deacylated tRNA. So what is the R state that RF2 reaches in the preHC? This is never actually addressed.

A larger question they might have asked is why not all bacteria have RF3, and whether this is related to a possible role of RF3 in quality control shown by the Green lab.

In summary, the paper deals with details of the mechanism of translational termination in bacteria that although important, will be of interest to only a handful of people even in the ribosome field. My own feeling is that it is not clear that it belongs in a general interest journal like *eLife*.

*Reviewer #3:*

The manuscript by Adio et al. describes an smFRET study of translation termination. Specifically, Adio et al. attempt to investigate whether RF2-meidated termination follows the same mechanism as RF1-mediated termination, the dynamics of the RF1-RF3-bound and RF2-RF3-bound termination complexes that are intermediates in the termination pathway, and the role of guanine nucleotide in RF3 function. The first two of these are open questions in the field and the third remains controversial, with two different models represented in the current literature. Given this, the answers to these questions would undoubtedly be of importance to the field and, in principle, the manuscript by Adio, et al. would be perfectly appropriate for publication in *eLife*.

However, as described in greater detail below, important controls are missing and, in several key instances, it is not clear that the data support the conclusions. In some ways, it almost seems as if the work presented here is a work in progress that is not yet finished and ready for publication. Thus, the authors would need to address these concerns before the appropriateness of this work for publication in *eLife* could be properly assessed.

1) The authors find that, unlike RF1, RF2 has a relatively low affinity for pre- and post-hydrolysis termination complexes and binds only transiently to these complexes. As the authors point out, this is a very surprising result. As such, it raises many important concerns that could be easily addressed by controls:

1a) The authors do not demonstrate whether their RF2 (or RF1) constructs are methylated at the Q of the GGQ motif in domain 3. Given that the Ehrenberg group has shown that the affinity of RF2 for termination complexes and the catalytic activity of RF2 on termination complexes are both dependent on this post-translational modification (Pavlov, et al. (1998) J Molec Biol and Dincbas-Renqvist, et al. (2000) EMBO J), it is important that the authors demonstrate that the surprising results they have obtained with RF2 are not due to the lack of this post-translational modification.

1b) The authors claim that their fluorophore-labeled RF2 (and RF1) construct are as active as their unlabeled counterparts, but the data are not shown. Given the results that the authors have obtained with their fluorophore-labeled RF2 construct, it seems to me that controls demonstrating that both the affinity and the catalytic activity of the authors' fluorophore-labeled RF2 construct are unchanged relative to unlabeled RF2 must be shown. It is also important to specify whether this comparison is being made to the unlabeled, single-cysteine mutant RF2 construct or to the unlabeled, fully wildtype RF2 construct. All of this is made more important by the fact that the authors have mutated and fluorophore-labeled a position on RF2, A237, that is very different from the position that they have mutated and labeled on RF1, has not been previously characterized, and is located in domain 3, where it could easily affect the affinity and/or catalytic activity of RF2 in a manner similar to that which is observed for the methylation of the Q in the GGQ motif in domain 3.

2) In several instances, the authors seem to interpret their data under the assumption that the termination complexes are saturated with unlabeled components, without having convincingly argued or demonstrated that the complexes are saturated:

2a) In the experiments in which unlabeled RF3 is added to termination complexes that are interacting with fluorophore-labeled RF2 shown in Figure 2—figure supplement Figure 2A, C. What is the affinity of the unlabeled RF3 for these complexes? Is the concentration of unlabeled RF3 that the authors use for these experiments high enough such that the complexes are saturated? How dependent are the interpretation of these data and the conclusions that are drawn on the complexes being saturated with unlabeled RF3? It seems like the extremely low, 10 nM concentrations of fluoropore-labeled RF2 and the possibility that, at any one time, the termination complexes are only partially occupied with RF3 would generate compositional heterogeneity that would make the data hard to interpret. The authors should address these questions through controls (e.g., titrations of fluorophore-labeled RF2 and/or unlabeled RF3) and/or revisions to the manuscript.

2b) Similar considerations apply to the interpretation of the experiments in which the authors characterize the dynamics of intersubunit rotation in the presence of unlabeled RF1 and RF3 or unlabeled RF2 and RF3. Particularly careful attention should be paid to the unlabeled RF2 and RF3 experiments, since the authors have discovered that RF2 binds to termination complexes with a very low affinity such that the termination complexes may not be saturated with RF2 at the RF2 concentrations that are used for these experiments. Such a scenario would again result in compositional heterogeneity that would make interpretation of the unlabeled RF2 and RF3 experiments difficult and, in addition, would challenge the appropriateness of comparing these results with the results of the RF1 and RF3 experiments in which the termination complexes are more likely to be saturated with RF1 (or at least have lower compositional heterogeneity due to the higher affinity of RF1 for termination complexes).

3) The authors need to be much more cautious regarding their assignment of the 0.5 FRET state that is observed in various intersubunit FRET experiments recorded in the presence of RF2. The authors have assigned this FRET state as corresponding to the rotated state of the ribosome and have made no distinction between this rotated state of the ribosome and the rotated state of the ribosome that is observed in other contexts (e.g., in the absence of any factors).

Nonetheless, as the authors and many others have pointed out, structural/steric considerations associated with intersubunit rotation make such an assignment very surprising. Given that it is based on a single FRET measurement on a single construct, how confident can the authors really be about this assignment? Is it possible that local conformational changes involving S6 and/or L9, but not associated with global rotation of the ribosome (or at least a full, global rotation of the ribosome) could lead to a decrease in the distance between the fluorophores so as to generate this decrease in FRET? What about photophysical considerations, could binding of RF2 have directly or indirectly altered the photophysical properties of one and/or the other fluorophore in a manner that is independent of intersubunit rotation? How do the authors' observation that RF2 has a low affinity for, and binds only transiently to, termination complexes play into this? Is it possible that sampling of the rotated state of the ribosome only happens under conditions in which RF2 has dissociated from the termination complex due to the low affinity (this relates to the concerns regarding whether the complexes are actually saturated with RF2)? Unless the authors can present arguments or controls to eliminate these alternative interpretations or, better yet, provide additional, independent data that RF2-bound termination complexes can occupy the rotated state, I don't think the assignment of this FRET state is supported by the data that has been presented here.

4) The fits to many of the dwell time histograms are biphasic, which indicates the presence of kinetic heterogeneity in the corresponding smFRET experiments. In each case, the authors should analyze the individual trajectories to determine and report whether a particular experiment exhibits static or dynamic heterogeneity and what the most likely origin of that heterogeneity is. The authors should be particularly attentive to static heterogeneity, which may be indicative of compositional heterogeneity arising from termination complexes that may not be saturated by a particular factor.

5) With the exception of Figure 6A, Figure 6B, and Figure 2—figure supplement Figure 2E, the data that are plotted and graphed do not have error bars. Additionally, the amplitudes and rate constants presented in Supplementary file 1 do not have standard deviations. Thus, it is not clear that the majority of the experiments were repeated and, if they were repeated, it is not clear why the authors have not performed and reported the statistical analyses necessary for assessing the reproducibility of the results and the validity of the interpretations. The authors should repeat the experiments and/or perform and report the statistical analyses of the data.

[Editors’ note: what now follows is the decision letter after the authors submitted for further consideration.]

Thank you for submitting your article "Dynamics of ribosomes and release factors during translation termination in *E. coli*" for consideration by *eLife*. Your article has been evaluated by James Manley (Senior Editor) and three reviewers, one of whom is a member of our Board of Reviewing Editors. The reviewers have opted to remain anonymous.

The reviewers have discussed the reviews with one another and the Reviewing Editor has drafted this decision to help you prepare a revised submission.

We have received comments from three reviewers (two of them new since the previous version). As you will read in the detailed comments, all three reviewers appreciated the substantial amount of work contained in the manuscript and the many interesting insights derived from the data. However, the reviewers remain concerned about the physiological relevance of the short peptidyl-tRNA ribosome complexes being studied. The authors offer cryoEM evidence that these tRNAs do readily sample the rotated state, especially at higher temperatures, and indeed, reviewer 2 suggests that there is additional literature supporting this point that should be clearly cited. Despite these arguments, the authors should acknowledge that these ribosome complexes carrying short peptidyl-tRNAs may not fully reflect the behavior of longer, more physiologically relevant, peptidyl-tRNAs where the classic ribosome configurations are more stabilized. As such, the authors should acknowledge that the detailed examinations here provide a starting point for defining the complex molecular events of termination rather than a definitive description.

In addition to these general concerns, reviewer 2 had numerous concerns about the statistical analysis throughout the study (in particular, whether the FRET histograms of the data subsets actually look like the FRET histograms of the total population). Reviewers 2 and 3 felt that conclusions were generally overstated given the limitations of the data (specifics are detailed in the reviews). Despite these limitations, all three reviewers felt that this manuscript contained important insights for the ribosome field, first on the relative differences in behavior between RF1 and RF2 on the ribosome (which is undoubtedly relevant to their different in vivo roles), and second, on the dynamics of RF1 and RF2 on the ribosome, and how they are impacted by the GTPase RF3. Finally, all three reviewers felt that the critical insights of the manuscript were often lost in the dense writing style, the overstatement of conclusions, and the lack of clarity in relating the work to some previous studies. At this stage, *eLife* will consider the manuscript for publication if the reviewer issues can be thoughtfully and completely addressed, both at the level of de-compressing the manuscript, clarifying the critical statistics, and more cautiously stating the conclusions.

*Reviewer #1:*

This revision of the manuscript by Adio et al. addresses most points raised by reviewers in the previous submission. All three reviewers were concerned about the peptidyl-tRNA migrating to the rotated state and the implications of this for the termination mechanism – the authors argue that this has been observed previously with relatively short peptidyl-tRNAs (they note in particular in cryoEM structures by Fischer et al.) and therefore rationalizes the data here showing rotation into the R state in complexes containing peptidyl-tRNAs. There were broad concerns related to the modification state of the RFs (whether fully methylated), the impact of the fluorescent labels on their behavior, the saturation of factors in various experiments, and the statistics of the analysis. The authors have systematically addressed each of these concerns to my satisfaction.

This paper represents a tour de force analysis with mountains of data (smFRET, ensemble FRET, and ensemble kinetic assays) on the conformational changes of various factors, tRNAs and the ribosome itself associated with peptide release and release factor binding (RF1, RF2, and RF3). These studies lead to several new findings that are important for defining the roles of these critical factors in translation termination, and indeed in defining the roles of such factors more broadly in biology. The most important findings are:

1) RF1 and RF2 behave quite distinctly on the preHC and postHC complexes – RF1 stabilizes the non-rotated state, RF2 enriches the rotated state and dissociates quickly (even without RF3). These differences are interesting in light of the auxiliary roles played by RF2 in quality control mechanisms in bacteria (both post-peptidyl QC as characterized by Zaher et al. and ArfA-mediated rescue). While the data don't tell us why these factors behave differently, they provide a biophysical basis for thinking about their distinct in vivo functions.

2) RF3-GTP stabilizes the rotated state and dissociates quickly from pre or post termination complexes.

3) Binding of RF3 to ribosomes bound with RF1 changes the conformation of both factors and stabilizes RF3 binding. RF1 and RF3 dissociation appears random; if RF3 dissociates first (from the non-rotated state), it does not require GTP hydrolysis, but if RF1 dissociates first, RF3 requires GTP hydrolysis to leave from the rotated state. It might be useful for the authors to compare the rates that they observe for RF1 departure as promoted by RF3 to previous studies determined by fluorescence (Koutmou et al.). Again, the differences here relative to RF2 (which does not depend on RF3 function) are interesting.

4) Rotation, peptide release, GTP hydrolysis are not coupled in a straightforward manner. These data are extremely dense, and include differences in behavior related to the type of GTP analog used (as previously reported in biochemical and structural studies). This section may have been the most difficult to sort through and I wonder whether the unnaturally short substrates (short peptidyl-tRNAs) and the lack of active release during the experiment (postHC complexes prepared by puromycin release) limited the impact/accuracy of the conclusions.

Overall, I feel the manuscript contains a substantial amount of important data on the dynamics and function of termination factors on the ribosome during translation termination. These data fit nicely with earlier studies by the same group detailing the critical role of the RF3-GTP cycle during these same steps (and extended here). The challenge for the manuscript remains that it is extremely dense and the main points are often lost in the detailed discussions of complex experiments. As just one example, the FRET distribution plots are layered with color (blue, pink and red, which all look very similar), to give the dimension of dynamics – which is useful and important, but nevertheless overwhelming. I broadly support publication of this work in *eLife* but would ask that the authors take one more pass to increase the accessibility of their main conclusions. Perhaps the problem is this: there are two stories here (1) the details of the functional cycle of RF1 and RF3 on the ribosome and (2) the distinctions in behavior between RF1 and RF2 on the ribosome. Yes, these are related stories, but presented together, the reader struggles to figure out whether to pay attention to commonalities or differences.

*Reviewer #2:*

The manuscript by Adio et al. describes an integration of single-molecule imaging and ensemble kinetic studies to explore the process of translation termination on the bacterial ribosome mediated by either release factor 1 or 2 (REF1, RF2) in concert with RF-3. The authors present a multitude of experiments describing the impacts of RF1 or RF2 binding on the conformational dynamics of various ribosome complexes, the rates of peptide release and the effects of RF3 on these various processes. Included in these investigations are direct measurements of factor binding interactions with the ribosome via FRET measurements as well as GTP hydrolysis studies to address an open question in the field about the role of GTP hydrolysis in the release mechanism and to refute reports that RF3-GDP is the physiological substrate for the ribosome in termination. There is no doubt that the synthesis of all of these data required tremendous effort both technically and intellectually. Although I did not see the original manuscript, it would appear that the addition of a second structural perspective on the classical-hybrid equilibrium (shown in Figure 3B, E) increase one's confidence in the interpretation that acylated tRNAs can indeed achieve hybrid-like configurations (see more about this below).

Although respectful of the amount of work that went into the present manuscript, my overarching conclusion is that it is exceedingly complicated. The salient physiologically relevant conclusions from the study are hard to grasp. The integration of ensemble and single-molecule experiments is of course extremely helpful at times as it provides confidence and grounding, but the number of experimental systems examined and the speculative conclusions made are dizzying, making it hard to keep track of key considerations. For instance, quantitative analyses are only provided for the subset of molecules that exhibit dynamics: it is not immediately clear how the proportion of non-dynamic/static molecules in each experiment affects the interpretations that are made; the existence of "static" and "dynamic" classes gives rise to a general concern about contributions of biochemical heterogeneities to the analyses presented. Do histograms of the small subset of dynamic molecules mirror the ensemble? The analyses presented are particular concerning given the authors use of/interpretation of these rate information, which is sometimes based on just 20-40 molecules from the hundreds that are measured (Figure 1A, B, E; Figure 5A; Figure 1—figure supplement 1A, C, E; Figure 1—figure supplement 2B, D; Figure 6—figure supplement 1A, C, D). Error bars on the individual measurements seem to be lacking throughout. The underlying basis of the static and dynamic populations is not clearly explained and should be clarified. As written, the manuscript seems to imply that this is expected from the biochemical system. But this is not clear to me where this notion comes from. The easier explanation is that this arises from rapid fluorophore photobleaching prior to evident conformational changes – in this context, I was unable to find the photobleaching rates for the distinct systems examined in the manuscript but it appears to be rapid (i.e. 0.5-1 per second) and thus a limiting feature to the experimental setup. Each of these concerns seem more or less consistent with the reviewer comments provided during initial review of the manuscript. My sense is that these considerations are likely to render the manuscript challenging to distill for the general reader.

One of the points raised by the initial reviewers is that significant complexities in the interpretation of the data presented arise from the use of ribosome complexes bearing short peptide mimics (fMET-Phe, fMET, NAc-Phe), which allows the ribosome to fluctuate between classical and hybrid states in the absence (and presence) of RFs. Although the authors choose to reference their own cryo-EM work indicating that the small subunit spontaneously and reversibly rotates at elevated temperatures, multiple studies prior and subsequent to their chosen reference have indicated hybrid-like states can be achieved with acylated-tRNA in the P site and that such hybrid-like configurations are sensitive to the nature and length of the nascent peptide (see Cornish et al., 2008; Cornish et al., 2009; Munro et al., EMBO 2008; Alejo et al., PNAS 2017). Referring to such states as "Hybrid" is inappropriate as structural data indicate that this includes A76 interactions with the C2394 region.

The average reader is likely to be confused by this potentially non-physiological aspect of the termination studies presented. As the initial reviewers suggested, had the study been performed with longer nascent chains, as is expected for the physiological substrate of release factors during translation termination, this complexity could likely have been entirely avoided. The fact that it is difficult to prepare such complexes (as the authors indicate in their response to reviewers) is fully appreciated, but a focused study on the propensity of the chosen complexes to achieve hybrid configurations and the nuanced impacts of such dynamics on release factor binding could be seen as an appropriate prerequisite for the present studies and their interpretation of the results presented.

For example, one of the key takeaways from the present study from the perspective of the general reader is that RF1 and RF2 perform differently in termination based on their measured off rates. Yet, such differences may simply arise from small distinctions in the binding energies of RF1 and RF2 to the ribosomes, which reveal themselves as potentially meaningful in the context of the non-physiological substrates examined. Such binding idiosyncrasies may be further exacerbated by the GGQ mutation that is used to prevent peptide release (as seems to be evident when comparing Figure 1—figure supplement 2C and E). What is the argument that these distinctions are physiologically important? Focused studies on the binding differences of RF1 and RF2 (concrete measurements of affinity, on and off rates, beyond the information that is presented) and clarification as to why such differences are important in the context of RF3 functions seems relevant to discuss but is presently lacking.

More readily addressable issues of concern include the following:

1) The use of language that could be considered inaccurate, misleading or too interpretive.

For instance, in the Introduction it states, "An alternative model was proposed, *when it turned out* that the affinity of RF3 to GDP and GDP[…]" Aren't these simply different measurements? As written it implies that the prior studies were just wrong and the studies by Koutmou and Peske are just right. How are the authors so sure that this is the case? Was the prior work retracted due to a technical error?

It is written in the Introduction, "thus the lifetime of the apo-RF3 complex would be too short to assume a tentative physiological role". The work cited seems to make a thermodynamic rather than a kinetic argument and the statement as written seems to create a false context.

In the Results its states that "the exact distribution of states depends on P-site tRNA […]" referencing Fei et al., 2011, when prior literature on the classical – to – hybrid equilibrium (N to R transition) were prior to report this finding. The authors should include references to Cornish et al., 2008; Cornish et al., 2009 and Munro et al., 2010 and Munro et al., 2010, which are already included in the bibliography.

The discussion in the subsection “RF1 and RF2 have distinct

effects on ribosome dynamics” regarding Figure 1—figure supplement 3 seem to suggest that there is information about factor binding in the data presented but this figure is confusing. The authors want to interpret differences between RF1 and RF2 'contour' plots that are post-synchronized as differences in binding and unbinding, but direct information about binding and unbinding is not present in these data. These types of plots may indicate differences in dynamics within the bound complexes and this needs to be clarified. While I have no doubt that there may be binding and unbinding information underlying the differences observed, the language used in this section is too strong. The authors should also try to put sufficient information in the figure legends to enable one to discern the relevant information about the experiment but looking at the figure legend alone. From the legend of Figure 1—figure supplement 3 it is impossible to know what ribosome complex was being examined (where does the FRET come from) without going back to the main text.

In the results it states that "the P/E hybrid state is short lived and decays rapidly (k_off_ = 6 per second)" As this is an experiment done in the presence of RF3, the rate constant k_off_ seems to imply unbinding of RF3, whereas I think the authors mean the classical to hybrid transition.

It also states that "suggesting the two processes are not tightly coupled, consistent with previous notions based on cryo-EM […] and smFRET work" While subtle, as stated this sentence implies that the cryoEM work came first and the smFRET work came later. But the opposite is true.

In reference to Figures 4C, G and K it is stated that "The FRET efficiency for the RF3-L11 pair is closer to 0.5, changed from 0.7, which is observed in the absence of RF1 (Figure 3C). Thus, the orientation of RF3 in the complex differs from that formed in the absence of RF1" While such conclusions may indeed be correct, statistical analyses on one dimensional histograms comprised of biological repeats are needed to make this conclusion without a second structural perspective to support this interpretation. The difference is not obvious and could be the result of day-to-day variabilities in microscope performance and/or fluorophore behaviors.

In reference to Figure 4 it is stated that "Thus, RF1 and RF3 can reside simultaneously on the Post HC in an arrangement where the position of RF1 and RF3 relative to L11 is shifted compared to the complex with only a single factor". Here, it seems too strong to make such statements without direct evidence of RF1 and RF3 binding.

*Reviewer #3:*

The manuscript by Adio et al. explores the mechanism of release factor recycling in bacteria using single-molecule FRET approach. The authors took advantage of different labeling strategies to look at ribosome, tRNA and release factor conformations at different stages of the termination process. These were used to draw two main conclusions: 1) Unlike RF1, RF2 can dissociate efficiently from the ribosome in the absence of RF3 (although RF3 helps); 2) The order of binding and dissociation events appear to be stochastic and unlike other translational GTPases, GTP hydrolysis by RF3 serves as a rescue mechanism for the factor.

Overall the paper was well written and the experiments appear to have been well executed. I was not however convinced that the paper offered significant new insights into the mechanism of translation termination. The notion that RF1 and RF2 are slightly different is not new (for example ArfA rescue of ribosomes only works in the presence of RF2 and not RF1). Furthermore the authors played down the effect of RF3 on RF2 (the fold change in termination rates under substoichiometric conditions is similar to that observed for RF1). Whether these differences also apply to other peptidyl tRNAs was not explored. As for the second main point of the paper, the Ruben group has also looked at the dynamics of the tRNAs and ribosome during recycling (albeit not to the same extent in this paper) and similar observations were made.

As for the presentation of the data, I felt that the results were over interpreted and the main conclusion was to a certain extent based on conjecture. For instance, the idea that the process of recycling is stochastic is based on the observation that many of the rates being similar (RF1, RF3 dissociation for example), but these experiments were conducted in the absence of peptide release (either using post hydrolysis complex or pre-hydrolysis one with GAQ RF1). It's unclear whether peptide release plays a role in the conformational changes. This is relevant, as under normal conditions release factors must bind in the presence of a peptidyl-tRNA and promote peptide release before it is recycled. At the end, I was left wondering how the different assays recapitulate what happens under normal conditions.

Another major point is the authors' assertion that GTP hydrolysis by RF3 merely serves to rescue the factor from nonproductive binding events. The data is based on the observation that in the presence of excess RF3, GTP hydrolysis in not required for RF1 turnover. The data in Figure 6A then suggests that RF3 is as likely to bind in a nonproductive fashion as to a productive one.

Given the overall contribution of the paper to the field together with the less than ideal interpretation of the data, my overall enthusiasm of the paper was tempered.

---

## [Author Response]

[Editors’ note: the author responses to the first round of peer review follow.]

Reviewer #1:[…] Concerns:In the first paragraph of the subsection “RF1 and RF2 have distinct effects on ribosome dynamics”, the authors state that the tRNA in the PreHC does not affect FRET dynamics (and for these experiments they use fMet-tRNA). However, the figures seem to depict a dipeptidyl-tRNA. This should be clarified in the text, figures, and figure captions.

We changed the figures to depict the formyl group by a different symbol than that representing an amino acid. The respective text has been changed/added (Figure 1 legend).

More importantly, it raises the question as to why they observe the rotated (R) state in the PreHC upon either RF2 or RF3 binding. Is this allowed rotation specific to the use of this minimal substrate fMet-tRNA? What would happen with a longer peptidyl-tRNA that extends into the exit tunnel? This is an important question because this finding suggests (contrary to the literature) that the hybrid tRNA states and rotated state of the ribosome can be separated.

Previous cryo-EM reconstructions have demonstrated that peptidyl-tRNA can assume a hybrid/rotated state (Fischer et al., 2010). There is a notion in the field that a peptidyl tRNA (and fMet-tRNA as a model peptidyl-tRNA) cannot move to the hybrid state. The cryo-EM work by Fischer et al. shows that this pertains to complexes prepared at low temperatures (4°C), but is not true for higher temperatures. At 18°C, the distribution of the subunit rotation states is bimodal, with the discernable peaks of non-rotated (about 60%) and rotated (40%) states, whereas at 37°C almost any rotation state is possible. Given that the smFRET data presented in this paper are obtained at 22°C, it is not surprising that fMet-tRNA can move into the rotated/hybrid state, which consistent with the cryoEM results (Figure 4 in Fischer et al., 2010).

To further validate the orientation of fMet-tRNA^fMet^ in the PreHC in the presence of RF2 or RF3 we performed additional experiments. We monitored the movement of fMet-tRNA^fMet^ into the hybrid state using FRET between the donor label at the tRNA (fMet-tRNA^fMet^(Cy3)) and the acceptor label on the ribosomal protein L1 (L1(Cy5)). This FRET pair has been extensively validated in previous studies as a reporter to probe whether the P-site tRNA is in the classical P/P state or in the hybrid P/E state (Munro et al., 2010a, Munro et al., 2010b, Munro et al., 2010c, Fei et al., 2008, Sternberg et al., 2009, Fei et al., 2009). We find that in the presence of RF2 the fraction of tRNAs in the P/E state is 20% which corresponds to the fraction ribosomes in the rotated state under the same conditions (Figure 3—figure supplement 1B). We also provide further controls as to the stability of fMet-tRNA^fMet^ in the presence of RF2(GAQ) mutant (Figure 1—figure supplement 2E). In the presence of RF3 the fraction of fMet-tRNA^fMet^ adopting the P/E state is almost 40% and the traces are extremely dynamic (now shown as Figure 3B, E). These findings further support our conclusion that RF3 changes the conformational dynamics of the PreHC. We also provide additional text explaining that peptidyl-tRNA can adopt a hybrid/rotated state (subsection “Interaction with RF3-GTP”).

Concerning the experiments with PreHC carrying a longer peptidyl chain, those are well beyond the scope of this paper for several reasons. First, fMet-tRNA^fMet^ has proven to be a good analog of peptidyltRNA with respect to studies of translation termination and several groups working on termination use it as a model system (Sternberg et al., 2009, Shi et al., 2016, Koutmou et al., 2014); the peptide length appears to have a small (if at all) effect the rate of peptidyl-tRNA hydrolysis (Indrisiunaite et al., 2015), which is not relevant for understanding the fundamental mechanism of termination. Second, the whole set of new experiments with a longer peptidyl-tRNA presents a significant technical challenge, because it requires preparation of homogeneous FRET-labeled ribosome complexes and validation of their biochemical and photophysical performance. Preparing such fully homogeneous complexes is challenging, because the efficiency of each translation step might be less than 100%; such technical problems cannot be solved within a reasonable time scale. Given that the hybrid/rotated state of peptidyl-tRNA has been reported for a peptidyl-tRNA (fMetVal-tRNA^Val)^by Fischer et al. (2010), and our data show that also fMet-tRNA^fMet^ can adopt a hybrid/rotated state, we feel that repeating the experiments with a longer peptidyl moiety would bring too little additional information to justify the enormous effort necessary to perform such additional experiments.

In Figure 3 and Figure 6G, release of RF3 is much faster than GTP hydrolysis, but one of the main conclusions is that RF3 requires GTP hydrolysis to be released from the rotated state (Figure 6D, H).

In the absence of RF1, RF3 can rapidly dissociate from some ribosomes, but these experiments do not show whether RF3 dissociates from the N or R state of the ribosome (Figure 3). In contrast, ensemble kinetics of Figure 6D shows that the back-rotation from the R state is very slow and the rate of backrotation corresponds to the rate of GTP hydrolysis (~0.5 s^-1^, Peske et al., 2014). Therefore, we suggest that the rapid dissociation monitored by smFRET reflects RF3–GTP dissociation from the N state, whereas RF3 release from the R state appears too slow to be captured by the smFRET technique. This is plausible, as the expected rate of GTP hydrolysis at smFRET conditions (22°C) is slower than 0.5 s^-1^, which is measured at 37°; the rates <0.2 s^-1^ are usually not resolved in our smFRET experiments. There is no contradiction between the data sets, as smFRET and ensemble measurements show different parts of the mechanism. In the presence of RF1, dissociation of RF3 is entirely independent of GTP hydrolysis. We changed the discussion in the subsection “The role of GTP binding and hydrolysis” to make this point clearer.

The confusion comes primarily from the RF3 dissociation experiments carried out with non-hydrolysable analogs, which do not behave as authentic GTP in our experiments. This is evident from experiments showing that RF3–GDPNP and RF3–GTPγS do not induce dynamic fluctuations of the ribosome–RF3 complex (Figure 6—figure supplement 1). If dynamic fluctuations are necessary for the rapid dissociation of RF3, and this is induced only by an authentic GTP-like conformation of RF3, then non-hydrolysable analogs are not suitable to study this question. These considerations motivated us to remove Figure 6G, H from the main text to the Figure 6—figure supplement 4. The text is modified to describe the potential problems of non-hydrolysable analogs.

Reviewer #2:[…] The implication of their control (with just tRNAs) is that the rotated state with RF1,2/3 is similar that reached with tRNAs during translocation. However, a P/E tRNA state is not possible for the preHC complex, because it requires a deacylated tRNA. So what is the R state that RF2 reaches in the preHC? This is never actually addressed.

The notion that the ribosomes with peptidyl-tRNA cannot adopt rotated/hybrid state is based on structural work (cryo-EM or X-ray) that typically used low temperatures for complex preparation, whereas our experiments are performed at 22°C. The results of cryo-EM studies (Fischer et al., 2010), who have extensively studied the distribution of the ribosome rotation states at different temperatures, show that peptidyl-tRNA (fMetVal-tRNA^Val^ in their experiments) favors the nonrotated/classical state only at low temperatures (4°C). In contrast, at 18°C the distribution of subunit rotation states is bimodal, with the discernable peaks of non-rotated (about 60%) and rotated (40%) states, whereas at 37°C almost any rotation state is possible (Figure 4 in Fischer et al., 2010). Given that the smFRET data presented in this manuscript are obtained at 22°C, it is thus not surprising that fMettRNA^fMet^ can move into the rotated/hybrid state, which is entirely consistent with the bimodal distribution revealed by the cryo-EM. We agree that this was not sufficiently explained in the manuscript and added the pertinent discussion in the revised manuscript.

Moreover, to address the question whether fMet-tRNA^fMet^ in PreHC can sample the P/E hybrid state we used FRET labels attached to the ribosomal protein L1 (L1(Cy5)) and fMet-tRNA^fMet^ (fMet-tRNA^fMet^(Cy3)). This FRET pair has been used in previous studies as a reporter to probe whether the P-site tRNA is in the classic P/P state or in the hybrid P/E state (Munro et al., 2010a, Munro et al., 2010b, Munro et al., 2010c, Fei et al., 2008, Sternberg et al., 2009, Fei et al., 2009). Our data clearly show that the presence of RF2 or RF3 not only promote subunit rotation but also the hybrid state formation of the tRNA (Figure 3C, F and Supplementary file 1).

A larger question they might have asked is why not all bacteria have RF3, and whether this is related to a possible role of RF3 in quality control shown by the Green lab.

This is an interesting question which is entirely beyond the scope of this paper. In *E. coli*, deletion of RF3 induces a higher expression of RF2 (Baggett et al., 2017). It is likely that bacteria that do not express RF3 compensate by having more RF2, or release RF1 with the help of other, yet uncharacterized factors (e.g. HflX).

In summary, the paper deals with details of the mechanism of translational termination in bacteria that although important, will be of interest to only a handful of people even in the ribosome field. My own feeling is that it is not clear that it belongs in a general interest journal like eLife.

On this point we politely disagree with the reviewer. We think that it is important to get the mechanism of translation termination right and make essential corrections to the inaccurate models that made their way to the textbooks. On a broader scale, this work shows that simple mechanistic models are not suitable to describe the dynamics of complex machineries, which may be an important lesson not only for the ribosome field but also for others dealing with macromolecular ensembles.

Reviewer #3:[…] 1) The authors find that, unlike RF1, RF2 has a relatively low affinity for pre- and post-hydrolysis termination complexes and binds only transiently to these complexes. As the authors point out, this is a very surprising result. As such, it raises many important concerns that could be easily addressed by controls:1a) The authors do not demonstrate whether their RF2 (or RF1) constructs are methylated at the Q of the GGQ motif in domain 3. Given that the Ehrenberg group has shown that the affinity of RF2 for termination complexes and the catalytic activity of RF2 on termination complexes are both dependent on this post-translational modification (Pavlov, et al. (1998) J Molec Biol and Dincbas-Renqvist, et al. (2000) EMBO J), it is important that the authors demonstrate that the surprising results they have obtained with RF2 are not due to the lack of this post-translational modification.

The factors are fully methylated. This is now shown in Figure 2—figure supplement 2 and stated in the respective text.

1b) The authors claim that their fluorophore-labeled RF2 (and RF1) construct are as active as their unlabeled counterparts, but the data are not shown. Given the results that the authors have obtained with their fluorophore-labeled RF2 construct, it seems to me that controls demonstrating that both the affinity and the catalytic activity of the authors' fluorophore-labeled RF2 construct are unchanged relative to unlabeled RF2 must be shown.

The activity is now documented in Figure 2—figure supplement 1.

It is also important to specify whether this comparison is being made to the unlabeled, single-cysteine mutant RF2 construct or to the unlabeled, fully wildtype RF2 construct.

The comparison is to the fully wild type unlabeled RF1 or RF2 (Figure 2—figure supplement 1).

All of this is made more important by the fact that the authors have mutated and fluorophore-labeled a position on RF2, A237, that is very different from the position that they have mutated and labeled on RF1, has not been previously characterized, and is located in domain 3, where it could easily affect the affinity and/or catalytic activity of RF2 in a manner similar to that which is observed for the methylation of the Q in the GGQ motif in domain 3.

The labeled position is C273, we apologize for the misprint. The activity of the factor is not changed (Figure 2—figure supplement 1).

2) In several instances, the authors seem to interpret their data under the assumption that the termination complexes are saturated with unlabeled components, without having convincingly argued or demonstrated that the complexes are saturated:2a) In the experiments in which unlabeled RF3 is added to termination complexes that are interacting with fluorophore-labeled RF2 shown in Figure 2—figure supplement 2A, C. What is the affinity of the unlabeled RF3 for these complexes? Is the concentration of unlabeled RF3 that the authors use for these experiments high enough such that the complexes are saturated? How dependent are the interpretation of these data and the conclusions that are drawn on the complexes being saturated with unlabeled RF3? It seems like the extremely low, 10 nM concentrations of fluoropore-labeled RF2 and the possibility that, at any one time, the termination complexes are only partially occupied with RF3 would generate compositional heterogeneity that would make the data hard to interpret. The authors should address these questions through controls (e.g., titrations of fluorophore-labeled RF2 and/or unlabeled RF3) and/or revisions to the manuscript.

The concentration of RF3 is 10-fold over the saturation concentration determined from the RF3-dependence of RF1 turnover rate (see Figure 2—figure supplement 1D and Zavialov et al., 2002). Despite this very large excess, it is difficult to entirely exclude a minor fraction of ribosomes that do not have RF3 at the moment when RF2 arrives. We would have easily recognized such complexes, because they are mostly static and favor the N state (Supplementary file 1). If some of the complexes have been taken for analysis, we would slightly underestimate the effect of RF3, which would make the observed trend even stronger.

2b) Similar considerations apply to the interpretation of the experiments in which the authors characterize the dynamics of intersubunit rotation in the presence of unlabeled RF1 and RF3 or unlabeled RF2 and RF3. Particularly careful attention should be paid to the unlabeled RF2 and RF3 experiments, since the authors have discovered that RF2 binds to termination complexes with a very low affinity such that the termination complexes may not be saturated with RF2 at the RF2 concentrations that are used for these experiments.

Despite the difference in the k_off_ values for RF1 and RF2, the complexes are saturated with factors at the high concentrations used in the inter subunit experiments (1 µM) (Figure 2—figure supplement 1B, C and Zavialov et al., 2002).

Such a scenario would again result in compositional heterogeneity that would make interpretation of the unlabeled RF2 and RF3 experiments difficult and, in addition, would challenge the appropriateness of comparing these results with the results of the RF1 and RF3 experiments in which the termination complexes are more likely to be saturated with RF1 (or at least have lower compositional heterogeneity due to the higher affinity of RF1 for termination complexes).

The ribosomes that do not have either RF1 or RF2 would show the rotation distribution of the complexes with RF3 alone. However, the distribution between the N and R states is clearly different with RF3 alone (73% R state) and RF3 with RF1 (43%) or RF3 and RF2 (56%). If some of the complexes lack RF1 or RF2, the effect would be underestimated, which does not change the trend, but makes it even stronger. We do not interpret small, statistically insignificant differences in rotation distribution between RF1+RF3 and RF2+RF3 experiments.

3) The authors need to be much more cautious regarding their assignment of the 0.5 FRET state that is observed in various intersubunit FRET experiments recorded in the presence of RF2. The authors have assigned this FRET state as corresponding to the rotated state of the ribosome and have made no distinction between this rotated state of the ribosome and the rotated state of the ribosome that is observed in other contexts (e.g., in the absence of any factors).Nonetheless, as the authors and many others have pointed out, structural/steric considerations associated with intersubunit rotation make such an assignment very surprising. Given that it is based on a single FRET measurement on a single construct, how confident can the authors really be about this assignment? Is it possible that local conformational changes involving S6 and/or L9, but not associated with global rotation of the ribosome (or at least a full, global rotation of the ribosome) could lead to a decrease in the distance between the fluorophores so as to generate this decrease in FRET? What about photophysical considerations, could binding of RF2 have directly or indirectly altered the photophysical properties of one and/or the other fluorophore in a manner that is independent of intersubunit rotation? How do the authors' observation that RF2 has a low affinity for, and binds only transiently to, termination complexes play into this? Is it possible that sampling of the rotated state of the ribosome only happens under conditions in which RF2 has dissociated from the termination complex due to the low affinity (this relates to the concerns regarding whether the complexes are actually saturated with RF2)? Unless the authors can present arguments or controls to eliminate these alternative interpretations or, better yet, provide additional, independent data that RF2-bound termination complexes can occupy the rotated state, I don't think the assignment of this FRET state is supported by the data that has been presented here.

The FRET pair used here to monitor subunit rotation is one of the best characterized FRET pairs (Sharma et al., 2016, Belardinelli et al., 2016, Quin et al., 2014, Ermolenko et al., 2007, Ermolenko et al., 2013, Cornish et al., 2008). It has been used to study subunit rotation on free ribosomes, diverse ribosome complexes with a number of different tRNAs and with and without elongation factors. Noller et al. and our group have shown that the S6-L9 FRET pair does not affect the ribosome function (Sharma et al., 2016, Belardinelli et al., 2016, Ermolenko et al., 2007, Ermolenko et al., 2013, Cornish et al., 2008, Hickerson et al., 2005, Majumdar et al., 2005). Clegg and Noller have tested all potential photophysical effects (Hickerson et al., 2005, Majumdar et al., 2005) and we have further characterized the behavior of the fluorophores upon peptide bond formation and translocation (Sharma et al., 2016, Belardinelli et al., 2016). We find the same two states – previously characterized as N and R – without the factors or with EF-G, EF-Tu, RF1, RF2, or RF3 regardless of the affinity of the factors. The FRET efficiencies are identical to those measured in other ribosome complexes and there is no peak broadening in FRET histograms. Termination factors cannot affect the reporter directly, because they bind from a different side of the ribosome, and there is no indication for the existence of some yet unobserved indirect effects. The available structures do not reveal anything unusual in the S6-L9 region upon termination factor binding; furthermore they demonstrate that the N state favors by RF1 or RF2 is the same N state as in the absence of the factors and likewise the R state formed with RF3 is similar to that formed during translocation. Thus, there is absolutely nothing to indicate that the two states we observe here are grossly different from the well-characterized N and R states observed so far with a wide variety of ribosome complexes.

Although there is no indication for the existing “alternative” rotation state, we made an additional experiment to show that the formation of the rotated state is accompanied by the tRNA movement into the hybrid state using another well-established FRET pair with the labels on the ribosomal protein L1 (L1(Cy5)) and fMet-tRNA^fMet^ (fMet-tRNA^fMet^(Cy3)). These experiments clearly show that ribosome rotation is accompanied by the hybrid state formation (Figure 3B, E); thus, two of the hallmarks of the R/hybrid H/classic states are satisfied. The law of parsimony does not leave us any other choice but to conclude that the N and R states formed in the presence of termination factors are the same as sampled by all other ribosome complexes studies so far. We assume that the attempt to re-interpret the R state observed in these experiments as an alternative conformation stems from the deeply-routed believe that a peptidyl-tRNA cannot move into the R/hybrid state. This is however true for low temperatures only, whereas at room temperature the complexes can adopt both N/classic and R/hybrid state (Fischer et al., 2010, Figure 4). (see also replies to reviewers 1 and 2).

4) The fits to many of the dwell time histograms are biphasic, which indicates the presence of kinetic heterogeneity in the corresponding smFRET experiments. In each case, the authors should analyze the individual trajectories to determine and report whether a particular experiment exhibits static or dynamic heterogeneity and what the most likely origin of that heterogeneity is. The authors should be particularly attentive to static heterogeneity, which may be indicative of compositional heterogeneity arising from termination complexes that may not be saturated by a particular factor.

We analyze each trace individually and we separate traces into static (no transitions) and dynamic traces (Supplementary file 1). We report (as indicated in Supplementary file 1) transition rates of dynamic traces only. There is no kinetic heterogeneity in the static traces. The decay rate of static traces has little biologic meaning as it is dominated by the photobleaching rate of the FRET dyes. The compositional heterogeneity is addressed in our replies above. The two rates obtained from two-exponential fitting are difficult to interpret, and in most cases we focus on the major (70-100%) component. In some cases, the heterogeneity has a biological meaning, which is discussed in the text.

5) With the exception of Figure 6A, Figure 6B, and Figure 2—figure supplement 2E, the data that are plotted and graphed do not have error bars. Additionally, the amplitudes and rate constants presented in Supplementary file 1 do not have standard deviations. Thus, it is not clear that the majority of the experiments were repeated and, if they were repeated, it is not clear why the authors have not performed and reported the statistical analyses necessary for assessing the reproducibility of the results and the validity of the interpretations. The authors should repeat the experiments and/or perform and report the statistical analyses of the data.

All statistics are now presented in Supplementary file 1.

[Editors' note: the author responses to the re-review follow.]

Reviewer #1:[…] This paper represents a tour de force analysis with mountains of data (smFRET, ensemble FRET, and ensemble kinetic assays) on the conformational changes of various factors, tRNAs and the ribosome itself associated with peptide release and release factor binding (RF1, RF2, and RF3). These studies lead to several new findings that are important for defining the roles of these critical factors in translation termination, and indeed in defining the roles of such factors more broadly in biology. The most important findings are:1) RF1 and RF2 behave quite distinctly on the preHC and postHC complexes – RF1 stabilizes the non-rotated state, RF2 enriches the rotated state and dissociates quickly (even without RF3). These differences are interesting in light of the auxiliary roles played by RF2 in quality control mechanisms in bacteria (both post-peptidyl QC as characterized by Zaher et al. and ArfA-mediated rescue). While the data don't tell us why these factors behave differently, they provide a biophysical basis for thinking about their distinct in vivo functions.

A notion on the non-canonical roles of RF2 has been included in the Introduction, first paragraph.

2) RF3-GTP stabilizes the rotated state and dissociates quickly from pre or post termination complexes.3) Binding of RF3 to ribosomes bound with RF1 changes the conformation of both factors and stabilizes RF3 binding. RF1 and RF3 dissociation appears random; if RF3 dissociates first (from the non-rotated state), it does not require GTP hydrolysis, but if RF1 dissociates first, RF3 requires GTP hydrolysis to leave from the rotated state. It might be useful for the authors to compare the rates that they observe for RF1 departure as promoted by RF3 to previous studies determined by fluorescence (Koutmou et al.). Again, the differences here relative to RF2 (which does not depend on RF3 function) are interesting.

The comparison with the rates reported in Koutmou et al. is included in the first paragraph of the subsection “Binding of RF1 and RF2 to the ribosome” and in the second paragraph of the subsection “Interplay between RF1 and RF3”. As stated in the text, we find that the data are in very good agreement given the different methods used to determine the dissociation rates.

4) Rotation, peptide release, GTP hydrolysis are not coupled in a straightforward manner. These data are extremely dense, and include differences in behavior related to the type of GTP analog used (as previously reported in biochemical and structural studies). This section may have been the most difficult to sort through and I wonder whether the unnaturally short substrates (short peptidyl-tRNAs) and the lack of active release during the experiment (postHC complexes prepared by puromycin release) limited the impact/accuracy of the conclusions.

We agree that the section on the role of GTP hydrolysis is complex. Also reviewer 2 commented that a combination of smFRET and ensemble kinetics is difficult to follow. To answer this criticism, we have restructured the text and removed the ensemble kinetics data. The arguments are now based on the smFRET data alone, which, we think, is easier to read (all ensemble and smFRET data give the same results anyway). We do not think that the short peptidyl substrates play a role here, as the key experiments on RF3 dissociation are carried out with PostHC where the peptide is released anyway. Furthermore, there is no indication that the puromycin treatment produces complexes that are different from those obtained with RF1, see the extended comparison between these complexes in Figure 4. The real surprise to us is why GTP hydrolysis is not needed for the recycling of the PostHC–RF1–RF3 complex, which is a naturally obtained Post complex with a deacylated tRNA in the P site.

Generally, we are surprised about the fundamental criticism concerning the short peptidyl-tRNAs as substrates, as this is a standard used by many in the termination field, e.g. Koutmou et al., 2014; Sternberg et al., 2009; Florin et al., 2017; Shi and Joseph, 2016; and a recent paper on rotation dynamics during termination by the Cornish lab (Casey et al., 2018). Ehrenberg and co-workers provided the direct comparison of the GTPase activity of RF3 on RF2-bound termination complexes with a tetrapeptide (MFTI) or fMet bound to P-site tRNA (Zavialov et al., 2003). They find that GTP hydrolysis rates are similar with MFTI and fMet which suggests that there is no significant difference in the interaction of termination factors with ribosomes carrying a short or a slightly longer peptide. Making the complexes with an authentic peptidyl-tRNA is not feasible, because the stop codons are read-through at the in vitro translation conditions in the absence of release factors and in the presence of large excesses of ternary complexes even at high accuracy conditions.

Overall, I feel the manuscript contains a substantial amount of important data on the dynamics and function of termination factors on the ribosome during translation termination. These data fit nicely with earlier studies by the same group detailing the critical role of the RF3-GTP cycle during these same steps (and extended here). The challenge for the manuscript remains that it is extremely dense and the main points are often lost in the detailed discussions of complex experiments. As just one example, the FRET distribution plots are layered with color (blue, pink and red, which all look very similar), to give the dimension of dynamics – which is useful and important, but nevertheless overwhelming. I broadly support publication of this work in eLife but would ask that the authors take one more pass to increase the accessibility of their main conclusions. Perhaps the problem is this: there are two stories here (1) the details of the functional cycle of RF1 and RF3 on the ribosome and (2) the distinctions in behavior between RF1 and RF2 on the ribosome. Yes, these are related stories, but presented together, the reader struggles to figure out whether to pay attention to commonalities or differences.

In order to make the representation of smFRET data less complex, we simplified the color scheme and now use only two colors (red and gray) to distinguish static from dynamic complexes. Furthermore, we removed the percentage of dynamic traces (% Dyn) from the histograms to simplify the graphs. We realized that as presented, this value apparently leads to confusion (see reply 1 to comments of reviewer 2). These values are still given in Supplementary file 1 where it is perhaps easier to understand. To make the writing style less dense, we extended the description of the experiments. Concerning the RF2 story, we are reluctant to separate it from the RF1/RF3 story, as they are intricately related. We hope that the general decompression of the paper helps to understand the results.

Reviewer #2:[…] Although respectful of the amount of work that went into the present manuscript, my overarching conclusion is that it is exceedingly complicated. The salient physiologically relevant conclusions from the study are hard to grasp. The integration of ensemble and single-molecule experiments is of course extremely helpful at times as it provides confidence and grounding, but the number of experimental systems examined and the speculative conclusions made are dizzying, making it hard to keep track of key considerations. For instance, quantitative analyses are only provided for the subset of molecules that exhibit dynamics: it is not immediately clear how the proportion of non-dynamic/static molecules in each experiment affects the interpretations that are made; the existence of "static" and "dynamic" classes gives rise to a general concern about contributions of biochemical heterogeneities to the analyses presented. Do histograms of the small subset of dynamic molecules mirror the ensemble? The analyses presented are particular concerning given the authors use of/interpretation of these rate information, which is sometimes based on just 20-40 molecules from the hundreds that are measured (Figure 1A, B, E; Figure 5A; Figure 1—figure supplement 1A, C, E; Figure 1—figure supplement 2B, D; Figure 6—figure supplement 1A, C, D).

The histograms displayed in the manuscript represent the entire population of molecules, rather than just a subset of dynamic traces. We realized that the “% Dyn” in the figures was probably misleading and removed it from the graphs. These values are still available in Supplementary file 1. The definition of static and dynamic traces is provided in the legend to Figure 1 and Supplementary file 1.

Error bars on the individual measurements seem to be lacking throughout.

Supplementary file 1 provides the detailed statistical analysis including the standard deviations of all measurements. It is impossible to provide statistical deviation in contour plots or time trajectories, but the information derived from these data (mean and sd of the FRET values, distribution of different FRET values in the population, and rate constants) is given either in the text, figure legends, or Supplementary file 1. As we now added the FRET distribution histograms (see below), we also included the mean ± sd for FRET values in figure legends (they are also available in Supplementary file 1). As indicated in figure legends and Materials and methods, standard deviations were calculated from three independent datasets. In figures and the calculations of statistics we followed the standards in the smFRET field (see Alejo et al., 2017, Ning et al., 2014; Wassermann et al., 2015; Elvekrog et al., 2013; Fei et al., 2011). We would be grateful to the reviewer for more specific comment as to which statistical parameter is still lacking.

The underlying basis of the static and dynamic populations is not clearly explained and should be clarified. As written, the manuscript seems to imply that this is expected from the biochemical system. But this is not clear to me where this notion comes from. The easier explanation is that this arises from rapid fluorophore photobleaching prior to evident conformational changes – in this context, I was unable to find the photobleaching rates for the distinct systems examined in the manuscript but it appears to be rapid (i.e. 0.5-1 per second) and thus a limiting feature to the experimental setup.

The existence of static and dynamic populations of different ribosome complexes has been consistently observed by many groups in the smFRET field, including the papers by Sternberg et al., 2009 and Casy et al., 2018 for the termination complexes and the groups of Gonzalez, Blanchard, Goldman/Cooperman, Cornish and us for other complexes during translation. The basis for the existence of static and dynamic populations is not known and the clarification of this point is beyond the scope of the manuscript. It is not a biochemical phenomenon related to the homogeneity of the PreHC or PostHC (which are tested and are uniform in all experiments), but it depends on the presence/absence of the peptide and on factor binding, in particular of RF3. Throughout the revised version, we simplified the description of ribosome fluctuations and removed the% Dyn which caused obvious confusion (see above). The stimulating effect of RF3–GTP on ribosome dynamics is shown in Figures 3 and 4 and described in the Results section.

The photobleaching rate is in the range of 0.07-0.19 s^-1^, very similar to photobleaching rates published by other groups (for example k_photobleaching_=0.05 ± 0.01 s^-1^-0.29 ± 0.03 s^-1^ in Sternberg et al., 2009). It is certainly much less than the 0.5-1 s^-1^ suggested by the reviewer, because if it were so high, we would not be able to measure dissociation rates as low as 0.2 s^-1^. The details of photobleaching estimations are described in Materials and methods (subsection “Data analysis”).

Each of these concerns seem more or less consistent with the reviewer comments provided during initial review of the manuscript. My sense is that these considerations are likely to render the manuscript challenging to distill for the general reader.

We made an utmost effort to satisfy the reviewers’ comments in the initial review. We included additional experiments including activity titrations of labeled termination factors with respect to unlabeled factors (Figure 2—figure supplement 1), provided mass spec analysis of labeled termination factors (Figure 2—figure supplement 2), and performed additional FRET measurements using dyes bound to tRNA^fMet^ and protein L1 (Figure 3B, E and Figure 3—figure supplement 1). The detailed statistics is provided in Supplementary file 1. We also rearranged the text and added citations to support our finding that termination complex can enter R/hybrid-like states in the presence of termination factors. Disregarding all this work of the 1^st^ revision does not seem fair, particularly for the issues of statistics and photobleaching (see answers above).

One of the points raised by the initial reviewers is that significant complexities in the interpretation of the data presented arise from the use of ribosome complexes bearing short peptide mimics (fMET-Phe, fMET, NAc-Phe), which allows the ribosome to fluctuate between classical and hybrid states in the absence (and presence) of RFs. Although the authors choose to reference their own cryo-EM work indicating that the small subunit spontaneously and reversibly rotates at elevated temperatures, multiple studies prior and subsequent to their chosen reference have indicated hybrid-like states can be achieved with acylated-tRNA in the P site and that such hybrid-like configurations are sensitive to the nature and length of the nascent peptide (see Cornish et al., 2008; Cornish et al., 2009; Munro et al., EMBO 2008; Alejo et al., PNAS 2017). Referring to such states as "Hybrid" is inappropriate as structural data indicate that this includes A76 interactions with the C2394 region.

We thank the reviewer for the support of the notion that short peptidyl-tRNAs can adopt an R state. One reason for citing the cryo-EM work is that it directly shows the distribution of the rotated states at elevated temperatures. We included the references suggested by the reviewer and used a more appropriate term “hybrid-like” instead of “hybrid”. We note, however, that we included those citations that report on subunit rotation with peptidyl-tRNA, rather than with deacylated tRNA (Munro et al., 2010) which is not disputed and is not relevant in the given context (subsection “RF1 and RF2 have distinct effects on ribosome dynamics”).

The average reader is likely to be confused by this potentially non-physiological aspect of the termination studies presented. As the initial reviewers suggested, had the study been performed with longer nascent chains, as is expected for the physiological substrate of release factors during translation termination, this complexity could likely have been entirely avoided. The fact that it is difficult to prepare such complexes (as the authors indicate in their response to reviewers) is fully appreciated, but a focused study on the propensity of the chosen complexes to achieve hybrid configurations and the nuanced impacts of such dynamics on release factor binding could be seen as an appropriate prerequisite for the present studies and their interpretation of the results presented.

The use of short model substrates is a standard approach in the field, as essentially every group working on termination uses this experimental system (Koutmou et al., 2014; Sternberg et al., 2009; Shi and Joseph, 2016; Casey et al., 2018). In that sense, we provide the data that can be compared with the bulk of previous work. Importantly, the current model of termination is based on very similar short peptidyl-tRNAs, so the ability to compare to the previous data is essential. There is no indication so far that these complexes are in any sense non-physiological. Pre-hydrolysis complexes with short peptidyl-tRNAs can bind termination factors, hydrolyze peptidyl-tRNA and proceed to recycling and thus the major features should represent termination complexes regardless of the peptidyl length. The potential effect of longer side chains for the dynamics of the PreHC is now indicated in the last paragraph of the Discussion.

Another question is which conclusions may depend on the hypothetical (albeit not demonstrated) effect of longer nascent chains. The dissociation rates of each factor alone (Figures 2 and 3) are the same on Pre and PostHC and thus an effect of a shorter or longer peptide chain is rather unlikely. In the presence of RF1 or RF2 the ribosomes favor the N state; this preference is unlikely to change with longer peptides, as it is assumed that they limit ribosome dynamics to the N state. The extent of ribosome rotation in the presence of RF3 may be affected and we added a sentence in the text to acknowledge this (Discussion), but RF3 dissociation rate is again independent of the Pre- or PostHC state, which makes the effect of chain length unlikely. For the experiments with RF1 and RF3 bound to the ribosome together, the nuances of rotation are not critical, as we interpret the tendencies by comparing the complexes with RF1 or RF3 alone with the complex with two factors. As to the PostHC, which constitute 2/3 of complexes used in this work, the peptide length is not important as it is already released. Thus, we think that the potential effect of the longer peptides is not fundamental, although we introduced the requested note of caution and the explanation as to why the work with natural long peptidyl-tRNA is not feasible (Discussion).

For example, one of the key takeaways from the present study from the perspective of the general reader is that RF1 and RF2 perform differently in termination based on their measured off rates. Yet, such differences may simply arise from small distinctions in the binding energies of RF1 and RF2 to the ribosomes, which reveal themselves as potentially meaningful in the context of the non-physiological substrates examined. Such binding idiosyncrasies may be further exacerbated by the GGQ mutation that is used to prevent peptide release (as seems to be evident when comparing Figure 1—figure supplement 2C and E). What is the argument that these distinctions are physiologically important? Focused studies on the binding differences of RF1 and RF2 (concrete measurements of affinity, on and off rates, beyond the information that is presented) and clarification as to why such differences are important in the context of RF3 functions seems relevant to discuss but is presently lacking.

Our experiments show clear differences in the binding properties of RF1 and RF2. We show that the factors differ in the way they affect the rotational dynamics of the ribosome and respond to the presence of RF3 and have very different residence times on the ribosome. This clearly shows that RF1 and RF2 interact with the termination complexes in a different way, which is consistent with crystal structures (Introduction, fourth paragraph). To make this statement, we compare three different conditions:

i) With RF1(GAQ) or RF2(GAQ) bound to PreHC where peptide release is prevented by mutation of the GGQ motif.

ii) With RF1 or RF2 bound to PostHC where the peptide is released using puromycin.

iii) With wt RF1 or RF2 with PostHC that is formed by natural peptide release.

The differences between the RF1 and RF2 (i.e. different residence times and different subunit rotation patterns) is obvious at all conditions tested. The finding that the dissociation rates of RF1 and RF2 differ also on PostHC (where no peptide is present at all) shows that the difference between RF1 and RF2 is not dependent on peptide length. The observed difference in the subunit rotation measured at saturating conditions of the factors (referred to in Figure 1) cannot be simply due to affinity differences between RF1 and RF2 because the complex is saturated with the respective factor (Figure 2—figure supplement 1).

Regarding the biochemical properties of RF1(GAQ) and RF2(GAQ) mutants, it is known that the affinity to the ribosome is not affected by the mutation (Zavialov et al., 2002). We have a full biochemical and ensemble kinetics analysis that shows that their binding properties are identical to those of the respective wild type proteins, but we are reluctant to include yet another layer of data into the manuscript, as all referees agreed that the manuscript is heavily loaded already. The fact that the GAQ mutants behave similarly to the wild type proteins on PostHC (Figure 1 and Figure 1—figure supplement 2) and that binding of RF3 to PostHC with RF1 or RF1(GAQ) results in identical changes in FRET value and dissociation rate (Figure 4) shows that the mutation does not induce a peculiar behavior of RF1.

More readily addressable issues of concern include the following:1) The use of language that could be considered inaccurate, misleading or too interpretive.For instance, in the Introduction it states, "An alternative model was proposed, when it turned out that the affinity of RF3 to GDP and GDP […]" Aren't these simply different measurements? As written it implies that the prior studies were just wrong and the studies by Koutmou and Peske are just right. How are the authors so sure that this is the case? Was the prior work retracted due to a technical error?

The K_D_ values presented by Zavialov et al., 2001 were obtained by nitrocellulose filtration quantifying the amount of radiolabeled nucleotide retained on the filter. NC filtration is a non-equilibrium method, which is sometimes unreliable and may underestimate the affinity for labile complexes. The papers that have used equilibrium methods (Koutmou et al., 2014 and Peske et al., 2014) provide more reliable estimates and explain the caveats of the nitrocellulose filtration; we do not think it is necessary to reiterate the published analysis. The issues with the NC filtration technique are also addressed in detail in Wilden et al., 2006 for EF-G. In that sense the choice of language was not interpretative, it summarized the existing publications. Nevertheless, we changed the sentence to avoid the statement disliked by the reviewer.

It is written in the Introduction, "thus the lifetime of the apo-RF3 complex would be too short to assume a tentative physiological role". The work cited seems to make a thermodynamic rather than a kinetic argument and the statement as written seems to create a false context.

The manuscript “Timing of GTP binding and hydrolysis by translation termination factor RF3” by Peske et al. studies both kinetic and thermodynamic aspects of nucleotide binding and exchange of RF3. Thus, there is no false context created, the argument is kinetic, as it is based on the reported rate constant of GTP binding to the nucleotide-free RF3.

In the Results its states that "the exact distribution of states depends on P-site tRNA" referencing Fei et al., 2011, when prior literature on the classical-to-hybrid equilibrium (N to R transition) were prior to report this finding. The authors should include references to Cornish et al., 2008; Cornish et al., 2009 and Munro et al., 2010 and Munro et al., 2010, which are already included in the bibliography.

We replaced Fei et al. with Cornish et al. 2008 which is the only paper addressing the extent of subunit rotation depending on the P site substrate.

The discussion in the subsection “RF1 and RF2 have distincteffects on ribosome dynamics” regarding Figure 1—figure supplement 3 seem to suggest that there is information about factor binding in the data presented but this figure is confusing. The authors want to interpret differences between RF1 and RF2 'contour' plots that are post-synchronized as differences in binding and unbinding, but direct information about binding and unbinding is not present in these data. These types of plots may indicate differences in dynamics within the bound complexes and this needs to be clarified. While I have no doubt that there may be binding and unbinding information underlying the differences observed, the language used in this section is too strong. The authors should also try to put sufficient information in the figure legends to enable one to discern the relevant information about the experiment but looking at the figure legend alone. From the legend of Figure 1—figure supplement 3 it is impossible to know what ribosome complex was being examined (where does the FRET come from) without going back to the main text.

We present rotation data alongside RF1/RF2 binding data (Figure 1—figure supplement 3, Figure 2). The two types of experiments complement each other and relate the rotation data to the presence of RF1/RF2 on the ribosome. Changes in subunit rotation provide a readout for factor binding. We rephrased the main text to a more careful statement. We extended the legend of Figure 1—figure supplement 3 such that the experimental procedure becomes clear.

In the results it states that "the P/E hybrid state is short lived and decays rapidly (k_off_ = 6 per second)…" As this is an experiment done in the presence of RF3, the rate constant k_off_ seems to imply unbinding of RF3, whereas I think the authors mean the classical to hybrid transition.It also states that "suggesting the two processes are not tightly coupled, consistent with previous notions based on cryo-EM[…] and smFRET work" While subtle, as stated this sentence implies that the cryoEM work came first and the smFRET work came later. But the opposite is true.

We changed the text und use k_closed→open_ to indicate transition rates between the P/P and P/E-like state. We also changed the order of cryo-EM and smFRET as pointed out by the reviewer.

In reference to Figures 4C, G and K it is stated that "The FRET efficiency for the RF3-L11 pair is closer to 0.5, changed from 0.7, which is observed in the absence of RF1 (Figure 3C). Thus, the orientation of RF3 in the complex differs from that formed in the absence of RF1" While such conclusions may indeed be correct, statistical analyses on one dimensional histograms comprised of biological repeats are needed to make this conclusion without a second structural perspective to support this interpretation. The difference is not obvious and could be the result of day-to-day variabilities in microscope performance and/or fluorophore behaviors.

We added one-dimensional histograms to illustrate FRET changes in response to factor binding together with mean FRET values ± sd in figure legends (Figure 2, 3, 4, 5, 6, Figure 1—figure supplement 3, Figure 6—figure supplement 2). Statistical analysis is also provided in Supplementary file 1. All histograms presented in this paper contain data from at least three individual datasets. Thus, day-to-day variabilities in microscope performance and/or fluorophore behaviors are already taken into consideration and integrated in the standard deviation of the measurements.

In reference to Figure 4 it is stated that "Thus, RF1 and RF3 can reside simultaneously on the Post HC in an arrangement where the position of RF1 and RF3 relative to L11 is shifted compared to the complex with only a single factor". Here, it seems too strong to make such statements without direct evidence of RF1 and RF3 binding.

We politely disagree. The biophysical properties (FRET efficiency with respect to L11 and the residence time) of RF1 and RF3 are very different when only one factor is bound compared to when they are present together. The change in the biophysical properties of RF1/RF3 can only result from the interaction with the respective other factor and therefore RF1 and RF3 must reside simultaneously on the ribosome. These changes can be used as a readout for factor binding, direct observation of both factors is not required to justify this statement.

Reviewer #3:The manuscript by Adio et al. explores the mechanism of release factor recycling in bacteria using single-molecule FRET approach. The authors took advantage of different labeling strategies to look at ribosome, tRNA and release factor conformations at different stages of the termination process. These were used to draw two main conclusions: 1) Unlike RF1, RF2 can dissociate efficiently from the ribosome in the absence of RF3 (although RF3 helps); 2) The order of binding and dissociation events appear to be stochastic and unlike other translational GTPases, GTP hydrolysis by RF3 serves as a rescue mechanism for the factor.Overall the paper was well written and the experiments appear to have been well executed. I was not however convinced that the paper offered significant new insights into the mechanism of translation termination. The notion that RF1 and RF2 are slightly different is not new (for example ArfA rescue of ribosomes only works in the presence of RF2 and not RF1).

The new mechanistic insights are the following:

1) RF1 and RF2 behave quite differently during canonical termination.

2) RF3–GTP can dissociate quickly from pre- or post-termination complexes without GTP hydrolysis.

3) Binding of RF3 to ribosome–RF1 complexes alters the positions of the two factors and stabilizes RF3 binding. The order of RF1 and RF3 dissociation appears random.

4) If RF1 is still present, dissociation of RF3 does not require GTP hydrolysis, but if RF1 dissociates first, RF3 dissociation is blocked if GTP hydrolysis cannot occur.

We suggest a model for translation termination which differs in all key points from current termination models and show how release factors affect each other’s function and ribosome dynamics.

Furthermore the authors played down the effect of RF3 on RF2 (the fold change in termination rates under substoichiometric conditions is similar to that observed for RF1). Whether these differences also apply to other peptidyl tRNAs was not explored.

We did not play down the effect of RF3 on RF2, as it is shown in Figure 2—figure supplement 3, where we report that even catalytic amounts of RF2 are sufficient to complete peptide release from PreHC. RF3 accelerates the reaction by a factor of 10. This effect is clearly much smaller than the effect of RF3 on RF1, which cannot recycle at all in the absence of RF3. Studying other tRNAs is well outside the scope of the paper, as other referees pointed out that it is already overloaded with experimental data.

As for the second main point of the paper, the Ruben group has also looked at the dynamics of the tRNAs and ribosome during recycling (albeit not to the same extent in this paper) and similar observations were made.

We refer to the work of the Gonzalez lab (Sternberg et al., 2009) on ribosome dynamics during termination. Indeed, some of the conclusions are in very good agreement with our results, such as the effect of RF3 on the L1 conformation of PostHC. However, Sternberg at al. did not compare the function of RF1 and RF2. Furthermore, they do not explicitly investigate the interaction of release factors with PreHC. As a result, they come to a different and less detailed mechanistic model. Our work complements the work presented by Sternberg et al. and provides new mechanistic insight into the importance of ribosome dynamics in general and in particular for the termination process.

As for the presentation of the data, I felt that the results were over interpreted and the main conclusion was to a certain extent based on conjecture. For instance, the idea that the process of recycling is stochastic is based on the observation that many of the rates being similar (RF1, RF3 dissociation for example), but these experiments were conducted in the absence of peptide release (either using post hydrolysis complex or pre-hydrolysis one with GAQ RF1). It's unclear whether peptide release plays a role in the conformational changes. This is relevant, as under normal conditions release factors must bind in the presence of a peptidyl-tRNA and promote peptide release before it is recycled. At the end, I was left wondering how the different assays recapitulate what happens under normal conditions.

The condition of active release is always included in our experiments.

To address the role of the nascent peptide for the dissociation of factors from termination complexes we compared three different conditions:

i) With RF1(GAQ) bound to PreHC where peptide release is prevented by mutation of the GGQ motif (Figure 1B-C, Figure 4A-D).

ii) With the same RF1(GAQ) bound to PostHC* where the peptide was pre-released using puromycin (Figure 1—figure supplement 2, Figure 4E-H).

iii) With wild type RF1 with PostHC that is formed by natural peptide release (Figure 1E-F, Figure 4I-L).

At all conditions, the rate of RF3 dissociation is the same, which shows that the interaction of RF3 with termination complexes in the presence of RF1 is independent of peptide release, subunit rotation and GTP hydrolysis. In contrast, RF1 dissociation is strongly dependent on peptide release. The rates of subunit rotation are faster than factor dissociation at all conditions. This highlights the stochastic aspect of RF3 interaction with termination complexes. Our model differs significantly from the linear termination models suggested so far and sheds light on how termination factors interact with each other and with the ribosome. We hope that there points became clearer in the revised version of the manuscript.

Another major point is the authors' assertion that GTP hydrolysis by RF3 merely serves to rescue the factor from nonproductive binding events. The data is based on the observation that in the presence of excess RF3, GTP hydrolysis in not required for RF1 turnover. The data in Figure 6A then suggests that RF3 is as likely to bind in a nonproductive fashion as to a productive one.

We changed Figure 6 and hope that this makes it clearer. Figure 6A and B show (biochemically) that RF1 recycling is blocked when GTP hydrolysis is abolished and RF3 is in sub-stoichiometric concentrations, but is efficient when RF3 is present in excess. We then show that RF3–GDPNP can bind to termination complexes with and without RF1 (Figure 6C, D). Finally, we show that the dissociation of RF3–GDPNP is slow in the absence and rapid in the presence of RF1. We simplified the text accordingly (subsection “The role of GTP binding and hydrolysis”) and added a more detailed discussion of these data in the Discussion. We never mention the term “non-productive binding” as we do not have an observable for that.

Given the overall contribution of the paper to the field together with the less than ideal interpretation of the data, my overall enthusiasm of the paper was tempered.

We hope that the text changes introduced upon revision have answered the main concerns of the reviewer and present our data and interpretation in a clearer way.